# NEURAL DAG SCHEDULING VIA ONE-SHOT PRIORITY SAMPLING

**Wonseok Jeon**[*], **Mukul Gagrani**[*], **Burak Bartan, Weiliang Will Zeng, Harris Teague**
**Piero Zappi, Christopher Lott**
Qualcomm AI Research[†]

## ABSTRACT

We consider the problem of scheduling operations/nodes, the dependency among which is characterized by a Directed Acyclic Graph (DAG). Due to its NP-hard nature, heuristic algorithms were traditionally used to acquire reasonably good solutions, and more recent works have proposed Machine Learning (ML) heuristics that can generalize to unseen graphs and outperform the non-ML heuristics. However, it is computationally costly to generate solutions using existing ML schedulers since they adopt the episodic reinforcement learning framework that necessitates multi-round neural network processing. We propose a novel ML scheduler that uses a *one-shot* neural network encoder to sample node priorities which are converted by list scheduling to the final schedules. Since the one-shot encoder can efficiently sample the priorities in parallel, our algorithm runs significantly faster than existing ML baselines and has comparable run time with the fast traditional heuristics. We empirically show that our algorithm generates better schedules than both non-neural and neural baselines across various real-world and synthetic scheduling tasks.

## 1 INTRODUCTION

The problem of scheduling operations arises across many domains, such as data centers where the incoming jobs have to be scheduled on a distributed server (Mao et al., 2019), manufacturing pipelines in the form of job shop scheduling problems (JSSP) (Manne, 1960), and ML compilers where the operations of a computation graph need to be scheduled on the available hardware devices (Paliwal et al., 2020; Zhou et al., 2020). In all these cases, the problem may be abstracted using a directed acyclic graph (DAG) where the nodes of the graph represent the operations and the edges represent the dependency constraints between the operations and hence the problem is also referred to as DAG scheduling. The objective is to minimize the finish time (or makespan) of the DAG subject to resource and dependency constraints.

It is well known that this is an NP-hard problem (Kan, 2012), and practitioners have traditionally relied on heuristic methods to obtain good solutions. One of the celebrated scheduling approaches is *list scheduling* (Graham, 1969) where the idea is to schedule nodes as early as possible and to break ties using priorities. The priorities can be obtained via different node metrics which are computationally inexpensive such as critical-path based, shortest processing time or most operations remaining (Haupt, 1989). More recently, researchers have proposed deep reinforcement learning based methods to solve scheduling problems (Zhang et al., 2020; Zhou et al., 2020; Wang et al., 2021; Mao et al., 2019). The scheduling policy in all the references utilize Graph Neural Networks (GNN) as an encoder to derive node embeddings. Zhang et al. (2020) proposed an auto-regressive GNN based policy for the JSSP problem which predicts the next node for scheduling given the nodes scheduled so far. Wang et al. (2021) proposed a bi-level optimization approach which modifies the input DAG by adding multiple edges via a learned policy and then apply the critical-path heuristic on the modified DAG. One major drawback of the existing ML based schedulers is the computational cost as they require multi-round neural network processing (encoding step). The multi-round neural network processing is reflected as auto-regressive architecture (Zhang et al., 2020) or bi-level optimization

---

[*]Equal contribution
[†]Qualcomm AI Research is an initiative of Qualcomm Technologies, Inc.

design (Wang et al., 2021). This drawback limits the scalability to large graphs and the applicability to domains where solutions need to be obtained in a timely manner (e.g., scheduling computation graphs in compilers).

In this paper, we propose a novel ML scheduler that uses a *one-shot* neural network encoder to sample node priorities which are converted by list scheduling to the final schedules. Since our encoder generates node priorities with a single forward pass of a neural network and efficiently samples priorities in parallel, our algorithm runs significantly faster than existing ML baselines and has comparable run time with the fast traditional heuristics.

The contributions of this paper are summarized below:

- We propose a novel end-to-end approach to learn scheduling priorities for list scheduling on DAGs. Our model adopts the recently proposed Topoformer architecture (Gagrani et al., 2022) as a DAG encoder and the Gumbel-Top-$k$ trick (Kool et al., 2019b) to sample node priorities (which are acquired by perturbing the encoder's output and converted into valid schedules via list scheduling). While optimizing our model with REINFORCE (Williams, 1992), we introduce logit norm regularization and cost standardization that significantly improve our model's representation power and performance compared to the model used in Gagrani et al. (2022).
- Our approach uses the one-shot encoder which generates the node priorities by running the Topoformer encoder once. This is in contrast of existing neural baselines (Wang et al., 2021; Zhang et al., 2020), all of which involves multi-round neural network processing. Due to the one-shot nature of our model, our method runs significantly faster than our neural baselines, while achieving runtimes slightly worse than yet comparable with those of computationally-efficient and simple non-ML heuristics.
- We show that our approach can be generally applied to a variety of scheduling tasks that includes JSSP, TPC-H benchmark and scheduling for synthetic and real-world computation graphs. For all benchmarks, our model outperforms both neural and non-neural baselines w.r.t. makespan metric (Wang et al., 2021; Zhang et al., 2020).

## 2 PRELIMINARIES

### 2.1 SCHEDULING PROBLEM

In scheduling problems, we define a DAG as a tuple $G := (\mathcal{V}, \mathcal{E}, \delta, \rho, \mu)$ with a set $\mathcal{V}$ of nodes (or vertices) and a set $\mathcal{E}$ of directed edges (or arcs). Each node $v \in \mathcal{V}$ represents an operation with $\delta(v) \geq 0$ denoting its operational duration and $\rho(v) \geq 0$ denoting the resources required to execute $v$. For a set $\mathcal{M}$ of machine types, each node $v \in \mathcal{V}$ has to be assigned to its own machine type $\mu(v) \in \mathcal{M}$ ($|\mathcal{M}| = 1$ and $|\mathcal{M}| > 1$ correspond to scheduling with homogeneous machines and heterogeneous ones, respectively). The set $\mathcal{E}$ of edges in the DAG $G$ represents computational dependency among nodes. For instance, for the scheduled start time $\tau(v) \geq 0, v \in \mathcal{V}$ for each node, a directed edge $(v_1, v_2) \in \mathcal{E}, v_1, v_2 \in \mathcal{V}$, means $\tau(v_1) + \delta(v_1) \leq \tau(v_2)$, i.e., any node should be scheduled on or after all its predecessor nodes are finished. We assume that each type of machine $m \in \mathcal{M}$ has its own maximum resource limit $\lambda(m) \geq 0$, i.e., at any point of time the total amount of occupied resources for machines of type $m$ cannot exceed $\lambda(m)$.

Let us introduce the vectorized notation $\tau = [\tau(v)]_{v \in \mathcal{V}} \in \mathbb{R}_{\geq 0}^{|\mathcal{V}|}$ of the start times with a little abuse of notation for the sake of simpler notation. We define a valid *schedule* as a vector $\tau \in \mathcal{T}$ where $\mathcal{T}$ is the set of all valid schedules (satisfying both precedence and resource constraints for given DAG $G$). The objective of the scheduling problem is to find $\tau^* := \arg\min_{\tau \in \mathcal{T}} C(\tau; G)$, where $C(\tau; G) := \max_{v \in \mathcal{V}}\{\tau(v) + \delta(v)\}$, the duration required to complete all operations, is the *makespan* of schedule $\tau$.

### 2.2 LIST SCHEDULING

List scheduling (Graham, 1969) is a class of priority-based schedule algorithms that are widely adopted in practice due to their simplicity. We describe how list scheduling works as follows:

**(Step 1)** Input a list of node priorities and set the current decision time to be zero.
**(Step 2)** Find *ready nodes* that can be scheduled at the current decision time, i.e., nodes whose predecessors have finished.

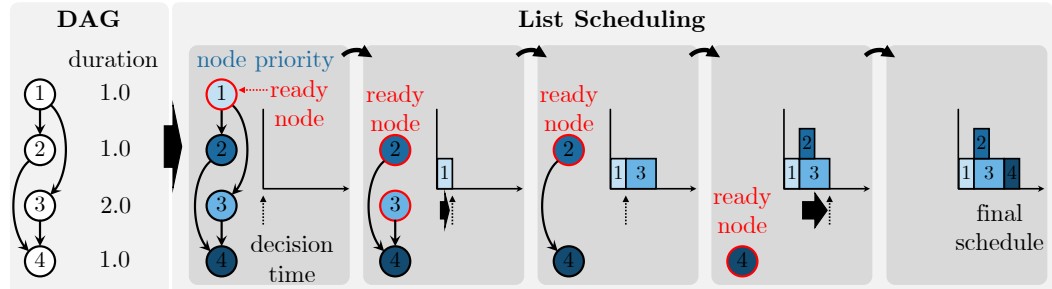

Figure 1: An example of list scheduling (Graham, 1969) for a 4-node DAG scheduling is described. Each node has its own duration, while resource limit is ignored for illustration purpose. List scheduling takes node priority as an input (e.g., $1 \succ 3 \succ 2 \succ 4$, brighter color implies higher priority) and schedules the higher-priority node among ready nodes earlier than the other nodes. After all ready nodes are scheduled, we move the decision time until a new set of ready nodes is found. We repeat these steps until we schedule all nodes.

**(Step 3)** Schedule the ready nodes sequentially at the current decision time by following the order of node priority until either all ready nodes are scheduled or further nodes cannot be scheduled due to resource constraints.

**(Step 4)** Move the decision time to the earliest finish time over all scheduled nodes which have not finished at the current decision time and repeat **(Step 2)** to **(Step 4)** until all nodes are scheduled.

We describe a simple example of list scheduling in Figure 1. Although Graham (1969) originally presented list scheduling for scheduling with homogeneous machines, we use the same definition of list scheduling for both homogeneous and heterogeneous machines.

### 2.3 THE GUMBEL-TOP-$k$ TRICK

Consider a random variable $Y$ over a set $\mathcal{Y}$ of finite categories, where the distribution is defined by the softmax over $\mathrm{logits}(y) \in \mathbb{R}, y \in \mathcal{Y}$ (the unnormalized log-probabilities), i.e., $\Pr\{Y = y\} \propto \exp(\mathrm{logits}(y)), y \in \mathcal{Y}$, and thus

$$\Pr\{Y = y\} = \frac{\exp(\mathrm{logits}(y))}{\sum_{y' \in \mathcal{Y}} \exp(\mathrm{logits}(y'))}. \tag{1}$$

The *Gumble-Max trick* (Gumbel, 1954) is a method to sample from the categorical distributions when $\mathrm{logits}$ characterizing the distributions are tractable. Specifically, the trick shows that by using a random vector $Z \in \mathbb{R}^{|\mathcal{Y}|}$ where elements $Z(y), y \in \mathcal{Y}$, are sampled from i.i.d. standard Gumbel distribution, one can randomly generate a category as follows:

$$\arg\max_{y \in \mathcal{Y}}\{\mathrm{logits}(y) + Z(y)\} \sim \Pr\{Y = y\}. \tag{2}$$

More recent works (Vieira, 2014; Kool et al., 2019b) found that the Gumbel-Max trick can be extended to *sample $k$ categories without replacement*, which is called the *Gumbel-Top-$k$ trick*. For the extension, they introduced $\arg\mathrm{top}^{(k)}$ which takes a real vector on $\mathcal{Y}$ and outputs a sequence of elements in $\mathcal{Y}$ that correspond to the $k$ largest values; the output sequence of the elements should be ordered by the corresponding decreasing input values (Kool et al., 2019b). As a special case where $k = |\mathcal{Y}|$, $\arg\mathrm{top}^{(k)}$ becomes $\arg\mathrm{sort}$ in decreasing values. The Gumbel-Top-$k$ trick generates the random sequence of elements in $Y$

$$[Y_1, Y_2, ..., Y_k] := \arg\mathrm{top}^{(k)}_{y \in \mathcal{Y}}\{\mathrm{logits}(y) + Z(y)\}, \tag{3}$$

and shows that the sequence is equivalent to the one from sampling $k$ elements without replacement; note that the random vector $Z$ is sampled *once* and *before* applying $\arg\mathrm{top}^{(k)}$. In other words, the distribution of the random sequence in Eq. (3) is shown to be described as follows (Kool et al., 2019b):

$$\Pr\{[Y_1, Y_2, ..., Y_k] = [y_1, y_2, ..., y_k]\} = \prod_{i=1}^{k} \frac{\exp(\mathrm{logits}(y_i))}{\sum_{y' \in \mathcal{Y} \setminus \{y_1, ..., y_{i-1}\}} \exp(\mathrm{logits}(y'))}. \tag{4}$$

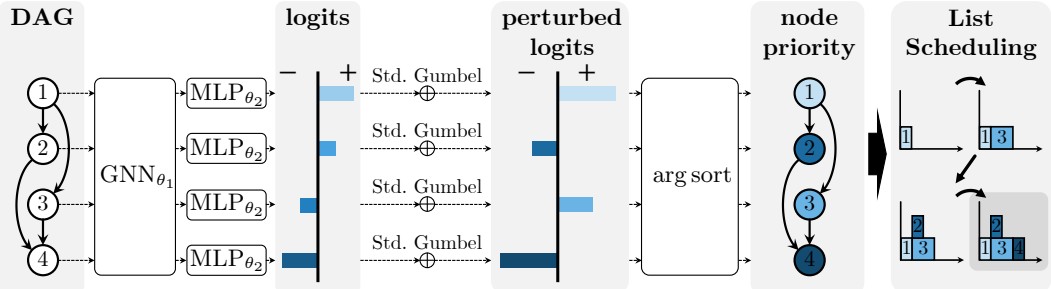

Figure 2: Our neural scheduler for DAGs works as follows: **(Step 1)** Generate logits for all nodes from a DAG by using a GNN encoder followed by an MLP. **(Step 2)** Perturb logits by adding i.i.d. Gumbel random variables. **(Step 3)** Take argsort over perturbed logits to define node priorities. Higher priority nodes have brighter colors. **(Step 4)** Use list scheduling to generate a schedule.

Intuitively, Eq. (4) tells us that each element in the random sequence in Eq. (3) follows the categorical distribution that is characterized by the softmax over logits, where previously sampled categories are excluded. In this work, we use Eq. (3) to decide the priorities over the elements in $\mathcal{Y}$ when $k = |\mathcal{Y}|$, which will be elaborated in the next section.

## 3 DAG SCHEDULING WITH NEURAL PRIORITY SAMPLER

We present our one-shot neural scheduler for DAG scheduling problems and its training method in this section. The content in this section is summarized as follows; we introduce the formal problem setting for the ML-based scheduling in Section 3.1; we describe how our model generates schedules by using the one-shot neural priority encoder and list scheduling in Section 3.2; the training method for our model and how it is relevant to the Gumbel Top-$k$ trick are discussed in Section 3.3 with some technical details to stabilize our algorithm, which we introduced in this work.

### 3.1 LEARNING-TO-SCHEDULE FRAMEWORK

Suppose we have a set $\mathcal{G} := \{G_1, G_2, ...\}$ of DAGs, where each DAG $G_i := (\mathcal{V}_i, \mathcal{E}_i, \delta_i, \rho_i, \mu_i)$ follows the definition in Section 2.1. We also assume that we have a device which is equipped with machines required for the DAGs. The learning-to-schedule algorithms by Zhang et al. (2020) and Wang et al. (2021) aim to find out a parameterized schedule generator $\pi_\theta(\tau|G)$ (with the neural network parameter $\theta$) that minimizes the average makespan over $\mathcal{G}$, i.e.,

$$\arg\min_\theta \mathbb{E}_{G\sim\mathcal{G}}\mathbb{E}_{\tau\sim\pi_\theta(\cdot|G)}\left[C(\tau; G)\right], \tag{5}$$

where $\tau$ is the schedule (the start times of nodes) and $C(\tau; G)$ is the makespan of $\tau$ in $G$ as in Section 2.1. The neural schedulers of existing works (Zhang et al., 2020; Wang et al., 2021) are sequential decision-making models that require multi-shot neural network processing. In contrast, we use a computationally efficient single-shot neural scheduler which is described in the next subsection.

### 3.2 SCHEDULE GENERATOR WITH ONE-SHOT PRIORITY SAMPLER

Using one-shot node priorities was recently proposed by Gagrani et al. (2022) to solve peak memory minimization problems in DAGs. Our neural scheduler is motivated by their idea and described in Figure 2. We adopt Topoformer, the graph neural network (GNN) encoder presented by Gagrani et al., as our graph encoder $\mathrm{GNN}_{\theta_1}(G) \in (\mathbb{R}^h)^{|\mathcal{V}|}$, where $h$ is the dimension of output embeddings for each node, and $\theta_1$ is the neural network parameter of the encoder. We use $\mathrm{MLP}_{\theta_2} : \mathbb{R}^h \to \mathbb{R}$ to convert the GNN's output node embeddings into *logits* over the nodes, i.e., for $\theta := (\theta_1, \theta_2)$ and $G \in \mathcal{G} = \{G_1, G_2, ...\}$,

$$\mathrm{logits}_\theta(v; G) := \mathrm{MLP}_{\theta_2}([\mathrm{GNN}_{\theta_1}(G)]_v) \in \mathbb{R}, v \in \mathcal{V}. \tag{6}$$

The difference between Gagrani et al.'s algorithm and ours arises from sampling procedure using logits. Gagrani et al. considers logits of schedulable nodes at each decoding step and sequentially

samples among those schedulable nodes, whereas we sample node priorities *only once at the start of decoding*. Specifically, by using i.i.d. standard Gumbel variables $Z(v) \in \mathbb{R}, v \in \mathcal{V}$, and $\arg\text{sort}$, we randomly sample a sequence of nodes from *perturbed logits*:

$$\vec{V} := [V_1, V_2, ..., V_{|\mathcal{V}|}] := \underset{v \in \mathcal{V}}{\arg\text{sort}}\{\underbrace{\text{logits}_\theta(v; G) + Z(v)}_{\text{perturbed logits}}\}. \tag{7}$$

Note that the LHS of Eq. (7) is a random sequence due to the randomness of $Z$ and $\arg\text{sort}$ is applied in decreasing values of the perturbed logits. We then regard the sampled random sequence $\vec{V} = [V_1, V_2, ..., V_{|\mathcal{V}|}]$ as the sequence of node priorities $V_1 \succ V_2 \succ ... \succ V_{|\mathcal{V}|}$, which does not require additional computation. Due to the stochastic nature of the Gumbel-Top-$k$ Trick described in Section 2.3, our mapping from the random sequence to the node priorities is equivalent to sampling nodes without replacement where *nodes sampled earlier are considered to be higher-priority ones*. More importantly, the trick allows us to use the tractable distribution of the random sequence, which becomes highly beneficial when optimizing the neural network. We will discuss this further in the next subsection.

Finally, we use list scheduling (described in Section 2.2) with the sampled random sequence in Eq. (7) to generate a valid schedule with the start time $\tau$:

$$\tau = \text{ListScheduling}(\vec{V}; G) \in \mathbb{R}_{\geq 0}^{|\mathcal{V}|}. \tag{8}$$

### 3.3 ALGORITHM AND PRACTICAL CONSIDERATION

The objective of learning-to-schedule frameworks in Eq. (5) can be rewritten with our scheduler generator as follows:

$$\underset{\theta}{\arg\min}\, \mathbb{E}_{G \sim \mathcal{G}} \mathbb{E}_{\vec{V} \sim \pi_\theta(\cdot | G)} C_{\text{LS}}(\vec{V}; G), \tag{9}$$

where $C_{\text{LS}}(\vec{V}; G) := C(\text{ListScheduling}(\vec{V}; G); G)$ is the makespan of list scheduling for given node priorities $\vec{V} = [V_1, ..., V_{|\mathcal{V}|}]$ and the graph $G$, and $\pi_\theta(\cdot | G)$ is the probability distribution of sampling node priorities for $G$. From the Gumbel-Top-$k$ trick discussed in Section 2.3, we can get the tractable form of $\pi_\theta(\cdot | G)$ below:

$$\pi_\theta([v_1, ..., v_{|\mathcal{V}|}] | G) := \text{Pr}_\theta\{\vec{V} = [v_1, ..., v_{|\mathcal{V}|}] | G\} = \prod_{i=1}^{|\mathcal{V}|} \frac{\exp(\text{logits}_\theta(v_i; G))}{\sum_{v \in \mathcal{V} \setminus \{v_1, ..., v_{i-1}\}} \exp(\text{logits}_\theta(v; G))}. \tag{10}$$

This enables us to use the following gradient descent rule using REINFORCE (Williams, 1992) that optimizes the objective Eq. (9) with learning rate $\alpha > 0$:

$$\theta \leftarrow \theta - \alpha \mathbb{E}_{G \sim \mathcal{G}} \mathbb{E}_{\vec{V} \sim \pi_\theta(\cdot | G)}[\nabla_\theta \log \pi_\theta(\vec{V} | G) C_{\text{LS}}(\vec{V}; G)]. \tag{11}$$

Together with the above update rule, we found that the practical techniques we introduce are crucial to stabilize the training and lead to significantly better performance, which are described below:

**Logit Norm Regularization.** When Gagrani et al. (2022) used their GNN encoder (Topoformer) to define the model distribution, the standardization over logits was used to bound the range of logits (See Section C.2 in Gagrani et al. (2022)). Specifically for the mean $m$ and standard deviation $s$ over $\text{logits}(v), v \in \mathcal{V}$, and a scalar hyperparameter $c > 0$, the standardized logits $\overline{\text{logits}}(v) := c \times (\text{logits}(n) - m)/s$ is used to define the model's probability of topological ordering. However, we empirically observe that such standardization over logits leads to poor performance. We believe this is because standardization significantly restricts the model's representation capability. While a rigorous proof for our argument may be complicated, we can easily show that this is true for binary random variables. Specifically with the above definition of standardized logits and a random variable $X \in \{0, 1\}$, one can only represent $(\text{Pr}\{X = 0\}, \text{Pr}\{X = 1\}) = (\frac{1}{1+\exp(2c)}, \frac{\exp(2c)}{1+\exp(2c)})$ or $(\frac{\exp(2c)}{1+\exp(2c)}, \frac{1}{1+\exp(2c)})$ (See Appendix A.). Note that the constant $c$ is a hyperparameter and assumed to be fixed during training, and the example implies that only two distributions can be described for any given $c$, which supports our claim.

---

**Algorithm 1** Neural DAG Scheduler via One-Shot Priority Sampling

---

**Input:** A set $\mathcal{G} = \{G_1, G_2, ...\}$ of the training graphs, a node priority sampler $\pi_\theta$, learning rate
   $\alpha > 0$, a regularization coefficient $c_{\text{logits}} > 0$.
 1: **for** each epoch **do**
 2:     **for** each $G \in \mathcal{G}$ **do**
 3:         Sample a batch of node priorities $\vec{V}^{(1)}, \vec{V}^{(2)}, ..., \vec{V}^{(N)} \sim \pi_\theta(\cdot|G)$.
 4:         Convert the priorities to valid schedules by list scheduling.
 5:         Evaluate makespan for all sampled schedules.
 6:         Standardize makespan $\bar{C}_n \leftarrow \overline{C}(\vec{V}^{(n)}; \vec{V}^{(1)}, ..., \vec{V}^{(N)}, G), n = 1, ..., N$, by Eq. (13).
 7:         Compute REINFORCE gradient $g_{\text{REINFORCE}} \leftarrow \frac{1}{N} \sum_{n=1}^{N} \nabla_\theta \log \pi_\theta(\vec{V}^{(n)}|G) \bar{C}_n$.
 8:         Compute logit norm gradient $g_{\text{logits}} \leftarrow c_{\text{logits}} \times \nabla_\theta L_{\text{logits}}(\theta; G)$ with Eq. (12).
 9:         Update $\theta$ by using gradient descent: $\theta \leftarrow \theta - \alpha(g_{\text{REINFORCE}} + g_{\text{logits}})$.
10:     **end for**
11: **end for**
**Output:** $\pi_\theta$

---

After removing the standardization of logits due to the above observation, however, we still observe instability in training caused by the unbounded logits. Therefore, we introduce the *logit norm regularizer* that minimizes

$$L_{\text{logits}}(\theta; G) := \frac{1}{|\mathcal{V}|} \sum_{v \in \mathcal{V}} \text{logits}_\theta(v; G)^2 \tag{12}$$

by gradient descent, together with the aforementioned REINFORCE objective. Intuitively, the norm regularizer allows our model to have the logits around the origin so that we can avoid numerical errors due to the unbounded logits while the sufficient amount of flexibility is still maintained for choosing logits. We empirically found that this highly stabilizes training and improves the performance and hence we apply the regularizer in all of our experiments.

**Cost Standardization.** Gagrani et al. (2022) stored the best-performing model so far and used its cost as the policy gradient baseline. This so-called greedy baseline was introduced due to the empirical performance of Kool et al. (2019a)'s algorithm on routing problems. However, if the scale of makespan varies significantly across different training graphs, the model trained with the greedy baseline can easily overfit a small subset of training graphs, which may lead to poor performance on test graphs. Also, the greedy baseline requires additional resources since the intermediate models should be stored and evaluated during training. To address these issues, we use cost standardization that has been widely adopted in policy-gradient algorithms, e.g., PPO (Schulman et al., 2017; 2015). Specifically for a given graph $G$ during a training iteration, we first sample multiple node priorities, i.e., $\vec{V}^{(1)}, ..., \vec{V}^{(N)} \sim \pi_\theta(\cdot|G)$, and evaluate makespan $C_{\text{LS}}(\vec{V}^{(1)}; G), ..., C_{\text{LS}}(\vec{V}^{(N)}; G)$. During the policy gradient update, we use the standardized makespan across the samples, i.e.,

$$\overline{C}(\vec{V}; \vec{V}^{(1)}, ..., \vec{V}^{(N)}, G) := \frac{C_{\text{LS}}(\vec{V}; G) - \text{mean}_{n=1,...,N}[C_{\text{LS}}(\vec{V}^{(n)}; G)]}{\max\{\text{std}_{n=1,...,N}[C_{\text{LS}}(\vec{V}^{(n)}; G)], \epsilon\}}, \tag{13}$$

where $\epsilon > 0$ is used to clip the standard deviation in denominator for numerical stability.

The final algorithm with additional stabilization ideas is summarized in **Algorithm 1**. We train our model with multiple training epochs over $\mathcal{G}$ where a single epoch considers each graph in $\mathcal{G}$ once. For each graph $G \in \mathcal{G}$, we randomly sample $N$ priorities and use REINFORCE to update the model parameter $\theta$ while regularizing the norm of logits by Eq. (12). We multiply with a constant $c_{\text{logits}} > 0$ the regularization loss to balance between REINFORCE loss and regularization. We set $c_{\text{logits}} = 0.001$ and observe that it empirically works well in all of our experiments.

## 4 RELATED WORKS

**ML for Combinatorial Optimization.** The idea of using ML to solve DAG scheduling fits into the broader theme of using ML for Combinatorial Optimization (CO) which has received attention recently (Bengio et al., 2021). Most of the works in the literature in ML for CO use RL to learn a

policy to select actions for reward maximization which is set to be a direct function of the problem objective. The policy can be an end-to-end policy whose actions correspond to the the solution of the CO problem (Zhou et al., 2020; Kool et al., 2019a; Joshi et al., 2021; Khalil et al., 2017; Zhang et al., 2020) or the policy can augment a traditional heuristic/solver of the problem to find better solutions (Paliwal et al., 2020; Xin et al., 2021; Ahn et al., 2020; Wang et al., 2021).

**End-to-End ML Schedulers.** In the context of the scheduling problems, Zhang et al. (2020) and Park et al. (2021) proposed an end-to-end GNN based policy to solve the JSSP problem which is a special case of DAG scheduling. Their policy is auto-regressive which selects the nodes to be scheduled iteratively and at each iteration they run the GNN on the modified disjunctive graph to get new node embeddings. Thus, they are required to run the GNN encoder $|N|$ times which is prohibitive for large graphs. In order to address the complexity of auto-regressive policies for large graphs encountered in compiler workflows, Zhou et al. (2020) came up with the idea of iterative refinement which refines the generated schedule by running their GNN policy $K$ times where $K$ is a hyper-parameter. Sun et al. (2021), Mao et al. (2019) and Zhou et al. (2022) consider the problem of scheduling jobs on data clusters and provide end to end deep RL solutions to solve data center scheduling under various settings.

**Hybrid Schedulers.** Wang et al. (2021) propose a bi-level optimization approach for DAG scheduling where they learn a policy which modifies the input DAG by adding edges and then use the critical-path based list scheduling method on the modified DAG to obtain a schedule for the original problem. The authors allow upto $K$ edges to be added to the DAG (where $K$ is a hyper-parameter) and their policy can run the GNN $K$ times to get to the final schedule. Paliwal et al. (2020) proposed a neural augmented genetic algorithm for scheduling in compilers. They used a GNN policy to learn the parameters of the mutant distribution which was used by the genetic algorithm to find good schedules.

In contrast to these works, our method generates the node priorities end-to-end requiring only a single pass of our GNN encoder and uses list scheduling to obtain the final schedule. This makes our approach more efficient and scalable compared to prior works.

## 5 EXPERIMENTS

We evaluate our neural DAG scheduler and baselines for three different scheduling tasks: **JSSP**, DAG scheduling on **TPC-H dataset**, and scheduling on **computation graphs** (See Appendix B for details about all tasks.). We compare our algorithm with both non-neural and neural baselines. For non-neural baselines, we consider list scheduling algorithms with different node priorities (based on Critical-Path **(CP)**, Most Operations Remaining **(MOPNR)** and Shortest Processing Time **(SPT)**) (Zhang et al., 2020), and a constraint programming **(Const. Prog.)** solver (CP-SAT by Perron & Furnon) with 24 hours time limit. For neural baselines, we first consider Learning-to-Dispatch **(L2D)** algorithm by Zhang et al. (2020) in the experiment with synthetic JSSP instances, where we train the deployed implementation using the dataset. We also consider **PPO-BiHyb** (Wang et al., 2021), where we use the deployed model of PPO-BiHyb for TPC-H dataset and train the model for the dataset with real-world computation graphs.

For our model, we consider two types of operating modes. In **Greedy** mode, our model generates schedules with node priorities that uses $\arg \text{sort}$ and the pure logits without adding Gumbel random variables. In **Sampling** mode, we sample multiple node priorities (where the number of samples are chosen from 16, 64 and 256, each of which corresponds to S(16), S(64), S(256) in Tables) and output the best schedule from the priorities. We report the makespan for output schedules and the run time of each algorithm. We report the speedup metric, which is defined as the ratio of sum duration of all nodes and the makespan, in the computation graph scheduling task. For all tasks and operating modes, we left the empirical result to show the effectiveness of our practical techniques in Appendix H.

### 5.1 JSSP

We evaluate our model on randomly generated JSSP instances from Zhang et al. (2020) which are defined by number of jobs $N_j$ and number of machines $N_m$. We summarize the results for $(N_j, N_m) = (25, 20), (25, 30), (50, 20)$ in Table 1. For each case, we train our algorithm for 100 training graphs and evaluate it for 50 unseen test graphs. We also train the neural baseline, L2D (Zhang et al., 2020), with the same training and test graphs. Note that this is different from the original

Table 1: Experiment results on synthetic JSSP instances are described. We use bold letters to emphasize the minimum average makespan for each JSSP instance.

| | $(N_j, N_m) = (25, 20)$ | | $(N_j, N_m) = (25, 30)$ | | $(N_j, N_m) = (50, 20)$ | |
| | Makespan | Time (sec) | Makespan | Time (sec) | Makespan | Time (sec) |
|---|---|---|---|---|---|---|
| CP | 2120.24 | 0.009 | 2588.72 | 0.017 | 3290.80 | 0.024 |
| SPT | 2265.88 | 0.002 | 2739.20 | 0.005 | 3548.20 | 0.008 |
| MOPNR | 2115.86 | 0.012 | 2625.96 | 0.023 | 3278.82 | 0.030 |
| L2D | 2253.94 | 1.245 | 2799.00 | 2.850 | 3452.70 | 3.711 |
| Greedy (ours) | 2032.70 | 0.021 | 2512.40 | 0.031 | 3108.56 | 0.049 |
| S(16) (ours) | 1970.98 | 0.054 | 2452.64 | 0.085 | 3032.44 | 0.138 |
| S(64) (ours) | 1948.76 | 0.127 | 2427.30 | 0.294 | 3009.08 | 0.469 |
| S(256) (ours) | **1932.42** | 0.514 | **2411.68** | 0.909 | **2997.10** | 1.527 |

Table 2: Experimental results on TPC-H datasets are described. We use bold letters to emphasize the minumim average makespan.

| | TPC-H-50 | | TPC-H-100 | | TPC-H-150 | |
| | Makespan | Time (sec) | Makespan | Time (sec) | Makespan | Time (sec) |
|---|---|---|---|---|---|---|
| Const. Prog. | **8629.4** | - | 19278.3 | - | - | - |
| CP | 9821.3 | 0.008 | 16914.1 | 0.027 | 24429.5 | 0.048 |
| SPT | 12818.4 | 0.002 | 19502.7 | 0.008 | 27409.4 | 0.021 |
| MOPNR | 11360.1 | 0.011 | 17733.1 | 0.032 | 24871.2 | 0.064 |
| PPO-BiHyb | 8905.4 | 66.484 | 15192.2 | 149.215 | 22371.2 | 571.424 |
| Greedy (ours) | 8845.6 | 0.057 | 14981.2 | 0.100 | 22332.7 | 0.259 |
| S(16) (ours) | 8782.4 | 0.114 | 14972.0 | 0.287 | 22330.2 | 0.674 |
| S(64) (ours) | 8742.5 | 0.216 | 14968.1 | 0.699 | 22323.0 | 1.856 |
| S(256) (ours) | 8694.4 | 0.540 | **14964.7** | 2.270 | **22320.8** | 6.485 |

training of L2D since L2D generates a new set of training graphs for every training iteration. The results show that our algorithm works well across different numbers of machines and also outperforms both neural and non-neural baselines. Due to one-shot decoding scheme in our model, the running time of our model is shorter than L2D. We observe that L2D ends up with worse performance than non-neural baselines although we train L2D until convergence. We think L2D does not generalize well with the limited number of training graphs, while our algorithm generalizes well with the same dataset.

## 5.2  TPC-H Dataset

TPC-H dataset [1] includes DAGs that consists of industrial queries and data modifications which represent computation jobs and need to be scheduled on a homogenous machine with finite resources. We use Wang et al.'s TPC-50/TPC-100/TPC-150 datasets for our experiments. Table 2 shows the performance of baselines and our model on the test set. We observe that our method obtains better average makespan than all the baselines including PPO-BiHyb on all the three instances except in TPC-50 where constraint programming achieves slightly better makespan. One more thing to note is that our method has much smaller run time compared to PPO-BiHyb. This is because our method generates node priorities via a single pass of our GNN encoder and samples priorities effectively by using Gumbel Top-$k$ trick, whereas PPO-BiHyb has to run their GNN encoder multiple times and requires beam search.

## 5.3  Computation Graphs

We test our approach on scheduling tasks for both synthetic and real-world computation graphs of neural networks that arise in ML compilers. We consider three type of synthetic computation graphs:

---

[1]http://tpc.org/tpch/default5.asp

Table 3: Experiment results on synthetic computation graphs are described. We use bold letters to emphasize the maximum average speedup for each graph distribution.

| | Layered Graph | | Erdos-Renyi | | Stoc. Block Model | |
|---|---|---|---|---|---|---|
| | SpeedUp | Time (sec) | SpeedUp | Time (sec) | SpeedUp | Time (sec) |
| CP | 4.580 | 0.058 | 5.049 | 0.055 | 4.701 | 0.055 |
| SPT | 4.526 | 0.013 | 4.541 | 0.008 | 4.473 | 0.007 |
| MOPNR | 4.745 | 0.075 | 5.112 | 0.068 | 4.761 | 0.068 |
| Greedy (ours) | 4.819 | 0.094 | 5.194 | 0.046 | 4.866 | 0.043 |
| S(16) (ours) | 4.848 | 0.311 | 5.211 | 0.214 | 4.916 | 0.198 |
| S(64) (ours) | 4.872 | 0.750 | 5.227 | 0.542 | 4.944 | 0.477 |
| S(256) (ours) | **4.889** | 2.418 | **5.239** | 1.799 | **4.965** | 1.533 |

Table 4: Experimental results on real-world computation graphs are described. We use bold letters to emphasize the maximum average speedup for each test set of graphs.

| | 200 - 500 Node Graphs | | 500 - 700 Node Graphs | | 700 - 1000 Node Graphs | |
|---|---|---|---|---|---|---|
| | SpeedUp | Time (sec) | SpeedUp | Time (sec) | SpeedUp | Time (sec) |
| Const. Prog. | 3.267 | - | 3.183 | - | 2.497 | - |
| CP | 3.174 | 0.007 | 2.804 | 0.016 | 2.739 | 0.025 |
| SPT | 3.107 | 0.002 | 2.868 | 0.005 | 2.664 | 0.008 |
| MOPNR | 3.181 | 0.009 | 2.825 | 0.020 | 2.739 | 0.028 |
| PPO-BiHyb | 3.223 | 17.937 | 2.965 | 52.777 | 2.798 | 322.793 |
| Greedy (ours) | 3.245 | 0.152 | 3.131 | 0.098 | 2.846 | 0.060 |
| S(16) (ours) | 3.271 | 0.192 | 3.188 | 0.245 | 2.848 | 0.230 |
| S(64) (ours) | 3.278 | 0.263 | 3.199 | 0.456 | 2.856 | 0.606 |
| S(256) (ours) | **3.286** | 0.595 | **3.207** | 1.309 | **2.860** | 2.001 |

layered graphs (Gagrani et al., 2022), Erdos-Renyi and stochastic block model graphs (Paliwal et al., 2020) (See appendix C.3.). Table 3 shows the performance of different methods on the synthetic computation graphs with 1000 nodes. We observe that our method outperforms all the traditional priority based baselines in terms of the achieved speedup (higher speedup implies lower makespan) for all three graph distributions. We also note that the run time of our method is competitive with that of the fast heuristics showing the scalability of our approach. Note that for scheduling applications in ML compilers, low run time of scheduler is crucial, and our results show the applicability of our model for such applications.

We also experiment on a set of proprietary real-world computation graphs to evaluate the practical applicability of our neural scheduler. This dataset consists of computation graphs of diverse neural net architectures like classifiers, convolution nets, denoisers, etc. We observe that our method achieves superior speedup on graphs of all sizes compared to all non-neural baselines. We also compare our approach with PPO-BiHyb and constraint programming baseline in this setting. Our results show that we outperform both PPO-BiHyb and constraint programming in terms of the achieved speedup. Note that we generate better schedules and run much faster than PPO-BiHyb. The fact that we beat constraint programming shows the large search space of this problem and the effectiveness of our approach to learn to find good schedules in this large space.

## 6  CONCLUSION AND FUTURE WORK

We propose an end-to-end approach to solve the general problem of scheduling over DAGs. Our method uses a single pass of Topoformer encoder to learn the node priorities which are used with list scheduling to generate a schedule. We also apply the Gumbel Top-$k$ trick to efficiently sample multiple node priorities and obtain better schedules. We demonstrate the effectiveness of our approach on a variety of tasks which include JSSP problems, TPC-H dataset and compiler scheduling datasets. We believe that our proposed approach can be extended to other CO problems such as routing problems (Kool et al., 2019a) and consider it a promising direction for future work.

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

## A  PROBABILITY DISTRIBUTION WITH STANDARDIZED LOGITS

Suppose our model has logits $l_0 := \text{logits}(0)$ and $l_1 := \text{logits}(1)$. One can easily show that the mean and standard deviation of logits are equal to $m = \frac{l_0 + l_1}{2}$ and $s = \sqrt{\frac{l_0^2 + l_1^2}{2} - m^2} = \frac{|l_0 - l_1|}{2}$, respectively. Now for the hyperparameter $c > 0$ that is assumed to be fixed, consider the standardized logits

$$\bar{l}_0 := \overline{\text{logits}}(0) := c \times \frac{l_0 - m}{s}, \bar{l}_1 := \overline{\text{logits}}(1) := c \times \frac{l_1 - m}{s} \tag{14}$$

and model a binary random variable $X \in \{0, 1\}$ by using them. The probability distribution using the softmax and the above standardized logits becomes

$$\Pr\{X = 1\} = \frac{\exp(\bar{l}_1)}{\exp(\bar{l}_0) + \exp(\bar{l}_1)} = \frac{1}{1 + \exp(\bar{l}_0 - \bar{l}_1)} = 1 - \Pr\{X = 0\}. \tag{15}$$

Since the difference between $\bar{l}_0$ and $\bar{l}_1$ is

$$\bar{l}_0 - \bar{l}_1 = c \times \frac{l_0 - m}{s} - c \times \frac{l_1 - m}{s} = c \times \frac{l_0 - l_1}{s} = 2c \times \frac{l_0 - l_1}{|l_0 - l_1|}, \tag{16}$$

we have

$$\Pr\{X = 1\} = \begin{cases} \frac{1}{1 + \exp(2c)}, & \text{if } l_0 > l_1, \\ \frac{\exp(2c)}{1 + \exp(2c)}, & \text{otherwise.} \end{cases}$$

and $\Pr\{X = 0\} = 1 - \Pr\{X = 1\}$.

## B  SCHEDULING TASKS

In this subsection, we describe how the different tasks that we consider fit into our scheduling framework described in section 2.1.

**Job Shop Scheduling Problem (JSSP).** JSSP is a special case of DAG scheduling where a set of $N_J$ jobs need to be scheduled on $N_M$ machines in $\mathcal{M}$. Each job consists of a sequence of operations that must go through $N_M$ machines in a specific order. This task can be modeled in the framework of section 2.1 by setting $\mathcal{M} = \{1, 2, \ldots, N_M\}$, we have one machine of each type i.e. $\lambda(m) = 1, \forall m \in \mathcal{M}$, each node has a machine type it can be scheduled on i.e. $\mu(v) \in \mathcal{M}$, each node occupies the machine it is scheduled on i.e. $\rho(v) = 1, \forall v \in \mathcal{V}$.

**DAG Scheduling on TPC-H.** In this task we need to schedule nodes with single machine type (homogeneous case) i.e. $\mathcal{M} = 0$ and all the nodes have same machine type i.e. $\mu(v) = 0, \forall v \in \mathcal{V}$. We assume the same setting as in Wang et al. (2021) for experiments and set the available resource of machine 0 to $\lambda(0) = 6000$. Node $v$ occupies $\rho(v) \in \mathbb{N}$ with $\rho(v) \leq 6000$ resources where $\mathbb{N}$ is the set of positive integers.

**Computation Graph Scheduling.** In this task we need to schedule computation operations of a DAG on a hardware with 3 types of machines so $\mathcal{M} = \{0, 1, 2\}$. We assume that we have 1 machine of type 0 and type 1 whereas 4 machines of type 2 i.e. $\lambda(0) = \lambda(1) = 1, \lambda(2) = 4$. This hardware setting is inspired from the structure of real ML accelerators which have multiple machines (threads) of different types available for data processing. A node $v$ can be scheduled on one machine of its machine type $\mu(v) \in \mathcal{M}$ i.e., the resource required $\rho(v) = 1, \forall v \in \mathcal{V}$.

## C  DATASET

### C.1  SYNTHETIC JSSP INSTANCES

We generate synthetic JSSP instances by using the code deployed by Zhang et al. (2020)[2]. For reproducibility of this dataset, we fix our seed as 0 and sample 150 instances for all combinations of $N_j \in \{25, 50\}, N_m \in \{10, 20, 30\}$, respectively. We use the first 100 instances as training instances and the last 50 instances as test instances.

---

[2]https://github.com/zcaicaros/L2D/blob/main/DataGen/generate_data.py

## C.2 TPC-H Dataset

We use TPC-H dataset generated by Wang et al. (2021)[3]. The dataset for each of TPC-50, TPC-100, TPC-150 experiments consists of 50 training graphs and 10 test graphs. The average number of nodes for TPC-50, TPC-100 and TPC-150 dataset are 467.2, 929.8 and 1384.5 respectively.

## C.3 Synthetic Computation Graph Dataset

**Layered graphs.** Layered graphs were introduced in Gagrani et al. (2022) as a model to generate synthetic graphs which have similar structure to the computation graph of a neural network. We use the default parameters specified in Gagrani et al. (2022) to generate the graphs. In order to generate the node duration, we first sample the memory size $m(v)$ of each node $v \in \mathcal{V}$ and then use the following affine model to generate the duration $\delta(v)$ for node $v \in \mathcal{V}$,

$$\delta(v) = \text{round}(100 \times m(v)) + 1. \tag{17}$$

We sample $m(v)$ by first sampling the value from Gaussian Mixture Model (GMM) $m(v) \sim \text{GMM}(\mathbf{w}, \mu, \sigma)\big|_{\mathbb{R}_+}$ and projecting it to non-negative values. We use a mixture of four Gaussians and set the means to $(\text{mean}_1, \text{mean}_2, \text{mean}_3, \text{mean}_4) = (0.5, 1, 3, 5)$ and standard deviations to $(\text{std}_1, \text{std}_2, \text{std}_3, \text{std}_4) = (0.5, 1, 1, 1)$.

We sample the machine type $\mu(v)$ of node $v$ such that $\Pr(\mu(v) = j) \propto \lambda(j)$ i.e. the probability that a node has machine type $j$ is proportional to the available resources $\lambda(j)$ for machine type $j$. In our computational graph experiments we used $\lambda(0) = \lambda(1) = 1, \lambda(2) = 4$ which leads to the following multi-nomial distribution for $\mu(v)$:

$$\Pr(\mu(v) = j) = \begin{cases} 1/6, & \text{if } j = 0, \\ 1/6, & \text{if } j = 1, \\ 2/3, & \text{if } j = 2. \end{cases} \tag{18}$$

**Erdos-Renyi and Stochastic Block Model.** These two graph distributions are well-known families of random undirected graphs and were used in Paliwal et al. (2020) for their experiments. We set the probability of an edge between any two nodes as $p = 0.05$ for the Erdos-Renyi graphs. We use the following parameters for the stochastic block model: the number of communities $k = 4$, the probability for an edge between two nodes of same community $p_{in} = 0.3$, the probability for an edge between two nodes of different communities $p_{out} = 0.001$. We use the procedure described in Paliwal et al. (2020) to convert the instances from these two graph families into a DAG. We use the same distribution to generate node duration and their machine type as in layered graph for these two graph distributions as well.

The results for 1000-node graphs are shown in the main part of our work, and 500-node graph results are in Appendix E.

## D Baseline Algorithms

### D.1 Priority-Based List Scheduling Baselines

We consider classical list scheduling baselines that use the following values as node priorities Zhang et al. (2020); Wang et al. (2021):

**Critical-Path (CP).** The priority for each node is determined by the length of the critical path from the node to the target node in a DAG.

**Most OPeratioNs Remaining (MOPNR).** The priority for each node is computed using the number of operations remaining on the longest path from the node to the target node in a DAG.

**Shortest Processing Time (SPT).** The priority for each node is computed using the inverse of the processing time of the node.

---

[3]`https://github.com/Thinklab-SJTU/PPO-BiHyb/tree/main/dag_data/tpch`

## D.2 Constraint Programming Baseline

Constraint programming models are commonly used to solve scheduling problems in the literature. We have developed constraint programming formulations and carried out numerical simulations to provide a baseline comparison for our methods.

1. **Constraint programming for computation graph scheduling.** The developed constraint programming problem has start times of jobs and machine assignments as discrete variables. The constraints consist of precedence constraints, correct machine assignment constraints, and constraints for enforcing no overlap of tasks if they are assigned to the same machine.
2. **Constraint programming for DAG scheduling with TPC-H dataset.** The constraint programming problem for the TPC-H dataset experiments has only the start times of jobs as variables. The constraints ensure that the precedence relations in a DAG are satisfied, and the total resource usage does not exceed the given resource budget. In addition, the task durations for this dataset have been scaled by 1000 and rounded to the nearest integer. The reason for this step is that the CP-SAT solver only works with integer valued data, and the duration values for this dataset are of floating point precision. We have solved the constraint programming problem using the scaled and rounded duration values. We have then divided the resulting makespan by 1000. This rounding step introduces an error of at most $0.0005$ seconds per node. Overall, for a 1000-node graph, this leads to an error of at most $0.5$ seconds in the worst case. This is negligibly small since the makespan values for this dataset are in the order of few thousand seconds.

We have implemented these formulations using Google OR-Tools and solved them using the CP-SAT solver (Perron & Furnon). Differently from other baselines, the constraint programming solver is guaranteed to find the optimal schedule given enough time. Due to practical reasons, we have set a time limit of 24 hours for the solver in the simulations. In most of the experiments, the time limit was reached before finding the optimal solution. In these cases, we have reported the best result obtained prior to the moment of time-out. Across all experiments, the solver has found the optimal solution for only 3 graphs in the TPC-H-50 dataset. For these 3 jobs, the amounts of time that the solver has run to find the optimal solutions are 2429, 21949, 39274 seconds, which are still significantly larger than our algorithms' run time.

We have observed that the memory requirements of the solver for large graphs could exceed the available memory in our servers. The graphs in TPC-H-150 fall into this category, where the experiments resulted in out-of-memory errors. We have not reported results for these graphs in Table 2.

In the experiments, we have provided initial feasible schedules for the solver. This is sometimes referred to as solution hinting or warm start. This speeds up the solver considerably. Initial feasible schedules could be obtained in different ways. One way is to generate a topological order of the DAG and map it to a sequential schedule without any parallelization. Another way is to run list scheduling and initialize the solver using the output of the list scheduling algorithm. We have experimented with both options. We have found that using the output of list scheduling leads to a better initialization for the solver.

## D.3 Neural Baselines

In this section we provide how our neural baselines are used. For all the tasks, we use a machine with a single GPU (Nvidia Tesla V-100) with 32 GB memory that is also used to train and evaluate our model.

**L2D.** While the original implementation of L2D uses a new set of training graphs for every training iteration, we consider more practical scenario where the number of training graphs is restricted. We use the aforementioned 100 training graphs and 50 test graphs for each JSSP instance. We use 4 environments (that was proposed by Zhang et al. (2020)), which ends up with training 4 graphs per training iteration. We train the L2D model for 3000 iterations over 5 different seeds and report the one that performs the best. Note that we tested learning rates $2 \times 10^{-5}$ (the default learning rate in their implementation) and $1 \times 10^{-4}$ for L2D and the learning rate $1 \times 10^{-4}$ works much better, and thus we report the results for this case.

Table 5: Experiment results on synthetic JSSP instances ($N_j = 25$).

| | $(N_j, N_m) = (25, 10)$ | | $(N_j, N_m) = (25, 20)$ | | $(N_j, N_m) = (25, 30)$ | |
| | Makespan | Time (sec) | Makespan | Time (sec) | Makespan | Time (sec) |
|---|---|---|---|---|---|---|
| CP | 1673.56 | 0.005 | 2120.24 | 0.009 | 2588.72 | 0.017 |
| SPT | 1807.90 | 0.001 | 2265.88 | 0.002 | 2739.20 | 0.005 |
| MOPNR | 1656.26 | 0.007 | 2115.86 | 0.012 | 2625.96 | 0.023 |
| L2D | 1725.58 | 0.577 | 2253.94 | 1.245 | 2799.00 | 2.850 |
| Greedy (ours) | 1578.58 | 0.018 | 2032.70 | 0.021 | 2512.40 | 0.031 |
| S(16) (ours) | 1540.92 | 0.045 | 1970.98 | 0.054 | 2452.64 | 0.085 |
| S(64) (ours) | 1527.40 | 0.092 | 1948.76 | 0.127 | 2427.30 | 0.294 |
| S(256) (ours) | **1519.16** | 0.321 | **1932.42** | 0.514 | **2411.68** | 0.909 |

Table 6: Experiment results on synthetic JSSP instances ($N_j = 50$).

| | $(N_j, N_m) = (50, 10)$ | | $(N_j, N_m) = (50, 20)$ | |
| | Makespan | Time (sec) | Makespan | Time (sec) |
|---|---|---|---|---|
| CP | 2903.14 | 0.012 | 3290.80 | 0.024 |
| SPT | 3111.74 | 0.003 | 3548.20 | 0.008 |
| MOPNR | 2897.60 | 0.016 | 3278.82 | 0.030 |
| L2D | 2964.96 | 1.084 | 3452.70 | 3.711 |
| Greedy (ours) | 2814.58 | 0.024 | 3108.56 | 0.049 |
| S(16) (ours) | 2800.32 | 0.064 | 3032.44 | 0.138 |
| S(64) (ours) | 2797.84 | 0.165 | 3009.08 | 0.469 |
| S(256) (ours) | **2796.98** | 0.644 | **2997.10** | 1.527 |

**PPO-BiHyb.** For the experiment with TPC-H dataset, we use the pretrained model[4] deployed by the author. We confirmed that the results reported in their paper are reproducible and evaluate the run time in our machine settings. For computation graph scheduling, we train PPO-BiHyb for 40 epochs (where 1 epoch is defined to train all training graphs in the dataset once).

# E  ADDITIONAL RESULTS

## E.1  JSSP

We describe empirical results for $(N_j, N_m) = (25, 10), (25, 20), (25, 30)$ in Table 5 and $(N_j, N_m) = (50, 10), (50, 20)$ in Table 6. For broader settings, we can see that our neural scheduler outperforms our baselines.

## E.2  COMPUTATION GRAPHS

We describe empirical results for both 500 and 1000 node synthetic computation graphs in Table 7. Over all sizes and graph distributions, our algorithm achieves better speedup within a short time.

# F  TRAINING AND MODEL DETAILS

## F.1  TRAINING

In this section we provide the training details of our model on the different scheduling tasks that we consider. For all the tasks we trained our model on a machine with a single GPU (Nvidia Tesla V-100) with 32 GB memory. We used Adam optimizer for training on all the tasks.

---

[4]https://github.com/Thinklab-SJTU/PPO-BiHyb/tree/main/pretrained

Table 7: Experiment results on synthetic graph datasets are described.

| | | Layered Graph | | Erdos-Renyi | | Stoc. Block Model | |
|---|---|---|---|---|---|---|---|
| | | SpeedUp | Time (sec) | SpeedUp | Time (sec) | SpeedUp | Time (sec) |
| 500 Node Graph | CP | 4.314 | 0.025 | 4.973 | 0.015 | 4.664 | 0.018 |
| | SPT | 4.214 | 0.004 | 4.480 | 0.002 | 4.416 | 0.003 |
| | MOPNR | 4.428 | 0.033 | 5.033 | 0.018 | 4.715 | 0.023 |
| | Greedy (ours) | 4.513 | 0.042 | 5.113 | 0.017 | 4.835 | 0.021 |
| | S(16) (ours) | 4.558 | 0.129 | 5.152 | 0.082 | 4.901 | 0.082 |
| | S(64) (ours) | 4.585 | 0.269 | 5.168 | 0.182 | 4.935 | 0.185 |
| | S(256) (ours) | **4.603** | 0.803 | **5.178** | 0.545 | **4.956** | 0.539 |
| 1000 Node Graph | CP | 4.580 | 0.058 | 5.049 | 0.055 | 4.701 | 0.055 |
| | SPT | 4.526 | 0.013 | 4.541 | 0.008 | 4.473 | 0.007 |
| | MOPNR | 4.745 | 0.075 | 5.112 | 0.068 | 4.761 | 0.068 |
| | Greedy (ours) | 4.819 | 0.094 | 5.194 | 0.046 | 4.866 | 0.043 |
| | S(16) (ours) | 4.848 | 0.311 | 5.211 | 0.214 | 4.916 | 0.198 |
| | S(64) (ours) | 4.872 | 0.750 | 5.227 | 0.542 | 4.944 | 0.477 |
| | S(256) (ours) | **4.889** | 2.418 | **5.239** | 1.799 | **4.965** | 1.533 |

**JSSP.** In this case we train our model on each JSSP instance with 100 training graphs and test it on 50 unseen graphs. We train our model for 20 epochs with 5 random seeds and pick the best performing model. We use the number of samples $N = 1000$.

**TPC-H Dataset.** We train our model on each TPC instance for 100 epochs with 10 random seeds and pick the best performing model. We use the number of samples $N = 2000$.

**Computation Graphs.** For each synthetic graph distribution, we consider the graph size equal to either 500 or 1000 and generate a training set of 3000 graphs and 300 unseen test graphs. For the real-world graphs, our dataset consists of 92 train graphs and 23 test graphs. We train for our model for 20 epochs with 5 random seeds and pick the best performing model. We use the number of samples $N = 1000$ for both synthetic and real computation graphs.

We summarize the time and epochs required for convergence in Table 8.

## F.2 Model architecture

We use topoformer encoder with the same hyperparameters as described in Gagrani et al. (2022). We use $c_{\text{logits}} = 0.001$ and $\epsilon = 0.1$ for clipping the denominator in Eq. (13).

## F.3 Input features

We use the following input node features $\vec{x}_v$ for node $v \in \mathcal{V}$:

- Node duration $\delta(v)$
- Node resource requirement $\rho(v)$
- One hot representation of node machine type $\mu(v)$
- The critical path duration from $v$ to the target node and critical path duration from source node to $v$

Note that for tasks where $\rho(v) = 1, \forall v$ we ignore $\rho(v)$ from the input feature. We normalize each entry $i$ of the input features across the nodes so that the features are in between 0 and 1 as follows:

$$\vec{x}_v[i] = \frac{\vec{x}_v[i]}{\max_{v' \in \mathcal{V}} \vec{x}_{v'}[i]}$$

In addition, we also augment the node features with the Laplacian positional encodings Dwivedi & Bresson (2020) of dimension 20 by computing it on the undirected version of the DAG. We pass the input feature $\vec{x}_v$ through a linear layer to obtain the initial embedding of node $v$ for the topoformer encoder.

Table 8: Training time details for the experiments. We report the rough duration for the convergence time for training and number of epochs to convergence for each of the experiment reported in the paper.

| Experiment name | Convergence time | Time per epoch | # epochs to converge |
|---|---|---|---|
| Real-world computation graphs | 25 min | 3 min | 8 |
| Syn. - Layered (500) | 2.5 hour | 1.2 hour | 2 |
| Syn. - Erdos-Renyi (500) | 3 hour | 1.2 hour | 2.5 |
| Syn. - Stoc. Block Model (500) | 3.5 hour | 1.2 hour | 3 |
| Syn. - Layered (1k) | 3 hour | 2.4 hour | 1.25 |
| Syn. - Erdos-Renyi (1k) | 8 hour | 2.4 hour | 3.3 |
| Syn. - Stoc. Block Model (1k) | 8 hour | 2.4 hour | 3.3 |
| TPC-H dataset - TPC-50 | 1 hour | 3.6 min | 16.7 |
| TPC-H dataset - TPC-100 | 5 hour | 14.4 min | 20.8 |
| TPC-H dataset - TPC-150 | 16 hour | 28.8 min | 33 |
| JSSP dataset - (25, 10) | 30 min | 2 min | 15 |
| JSSP dataset - (25, 20) | 45 min | 4.5 min | 10 |
| JSSP dataset - (25, 30) | 1 hour | 7.5 min | 8 |
| JSSP dataset - (50, 10) | 45 min | 4.5 min | 10 |
| JSSP dataset - (50, 20) | 3 hour | 12 min | 15 |

## G   PERFORMANCE IMPROVEMENT RELATIVE TO GAGRANI ET AL. (2022)

We use Topoformer (Gagrani et al., 2022) in our method and summarize the key differences between Gagrani et al. (2022)'s method and ours below:

- Our method
    1. Aims to solve scheduling problems
    2. Perturbs logits (via i.i.d. Gumbel noise) to sample node priorities (not necessarily satisfying precedence constraints) and convert the priorities into schedules via list scheduling.
    3. While using REINFORCE, cost standardization is used.
    4. Logits are regularized, i.e., its L2 norm is minimized together with REINFORCE loss.
- Gagrani et al. (2022)'s method
    1. Aim to solve peak memory minimization (which is *not* a scheduling problem).
    2. Logits are used to sample sequences *satisfying* the precedence constraints on DAGs.
    3. While using REINFORCE, greedy baseline is used, motivated by [15].
    4. Logits are standardized.

For items 3 and 4, our motivations are described as follows. REINFORCE with greedy baseline in Gagrani et al. (2022) uses the policy gradient

$$\mathbb{E}_{\vec{V} \sim \pi_\theta(\vec{V}|G)} \nabla_\theta \log \pi_\theta(\cdot|G)[C(\vec{V}; G) - C(\pi_{\theta'}(G); G)]$$

for a fixed $G$, where $\pi_{\theta'}$ is the greedy policy and its parameter $\theta'$ is copied from $\theta$ when the policy $\pi_\theta$ with $\theta$ shows the best performance during training (normally evaluated at the end of multiple epochs). The problems of greedy baseline are:

1. it may become unstable if cost scales differ too much across training graphs,
2. it slows down training since further forward computation with $\pi_{\theta'}$ is required,
3. it requires additionally memory to store $\theta'$,

all of which can be resolved by the cost standardization technique that we used in our paper. Also, if we use logit standardization used by Gagrani et al. (2022), we empirically observed performance degradation, which we believe is due to logit standardization restricting the representation power of the model and motivated us to use logit norm regularization instead (example in Appendix A).

To prove the effectiveness of our technical improvements, we did ablation studies for our algorithm with all tasks in our submission. Specifically, we compare our method with Gagrani et al. (2022)'s method using both greedy baseline and logit standardization while maintaining the priority sampling

Table 9: Experiment results on synthetic JSSP instances with ablation studies. Greedy and Sampling methods with (-) indicate results from the model trained with greedy baseline and logit standardization in Gagrani et al. (2022).

| | $(N_j, N_m) = (25, 20)$ | | $(N_j, N_m) = (25, 30)$ | | $(N_j, N_m) = (50, 20)$ | |
| | Makespan | Time (sec) | Makespan | Time (sec) | Makespan | Time (sec) |
|---|---|---|---|---|---|---|
| CP | 2120.24 | 0.009 | 2588.72 | 0.017 | 3290.80 | 0.024 |
| SPT | 2265.88 | 0.002 | 2739.20 | 0.005 | 3548.20 | 0.008 |
| MOPNR | 2115.86 | 0.012 | 2625.96 | 0.023 | 3278.82 | 0.030 |
| L2D | 2253.94 | 1.245 | 2799.00 | 2.850 | 3452.70 | 3.711 |
| Greedy (-) | 2077.60 | 0.022 | 2610.86 | 0.028 | 3129.16 | 0.056 |
| S(16) (-) | 2004.78 | 0.055 | 2493.38 | 0.083 | 3055.20 | 0.142 |
| S(64) (-) | 1979.86 | 0.129 | 2467.00 | 0.291 | 3037.38 | 0.484 |
| S(256) (-) | 1957.76 | 0.512 | 2446.04 | 0.844 | 3015.78 | 1.568 |
| Greedy (ours) | 2032.70 | 0.021 | 2512.40 | 0.031 | 3108.56 | 0.049 |
| S(16) (ours) | 1970.98 | 0.054 | 2452.64 | 0.085 | 3032.44 | 0.138 |
| S(64) (ours) | 1948.76 | 0.127 | 2427.30 | 0.294 | 3009.08 | 0.469 |
| S(256) (ours) | **1932.42** | 0.514 | **2411.68** | 0.909 | **2997.10** | 1.527 |

via Gumbel and list scheduling (this is necessary since we have to fairly compare performances in scheduling domains). We train the model for all scheduling tasks; JSSP, DAG scheduling, real and synthetic computation graphs.

### G.1 JSSP

Our method is shown to perform better than Gagrani et al. (2022)'s method (See Table 9.). Also, note that our method always outperform Gagrani et al. (2022)'s method when they are in the same mode and the same task. Runtimes for both methods are comparable with each other. We observe that Gagrani et al. (2022)'s method can outperform our neural and non-neural baselines for our JSSP instances.

### G.2 DAG SCHEDULING ON TPC-H DATASET

Our method outperforms Gagrani et al. (2022)'s method (See Table 10.). Runtimes for both methods are comparable with each other. For TPC dataset, our method can be regarded to make significant improvement due to the following observations:

1. S (256) of Gagrani et al. (2022)'s method (which shows the lowest makespan among the runs with Gagrani et al. (2022)'s method) is always outperformed by Greedy mode of our method (which is the highest makespan among our methods).
2. Our neural baseline, PPO-BiHyb Wang et al. (2021), always outperforms Gagrani et al. (2022)'s method for all TPC tasks.

### G.3 SCHEDULING COMPUTATIONAL GRAPHS

### G.3.1 SYNTHETIC GRAPHS

Our method outperforms Gagrani et al. (2022)'s method (See Table 11.). Runtimes for both methods are comparable with each other. For synthetic computation graphs, the following observations make our method more outstanding:

1. S (256) mode of Gagrani et al. (2022)'s method (that shows the lowest makespan among the runs using Gagrani et al. (2022)'s method) is always outperformed by Greedy mode of our method (which is the highest makespan among our methods).
2. Our non-neural baseline, MOPNR, always outperforms Gagrani et al. (2022)'s Greedy mode and some of sampling modes.

Table 10: Experiment results on TPC-H dataset with ablation studies. Greedy and Sampling methods with (-) indicate results from the model trained with greedy baseline and logit standardization in Gagrani et al. (2022).

|  | TPC-H-50 | | TPC-H-100 | | TPC-H-150 | |
|---|---|---|---|---|---|---|
|  | Makespan | Time (sec) | Makespan | Time (sec) | Makespan | Time (sec) |
| Const. Prog. | **8629.4** | - | 19278.3 | - | - | - |
| CP | 9821.3 | 0.008 | 16914.1 | 0.027 | 24429.5 | 0.048 |
| SPT | 12818.4 | 0.002 | 19502.7 | 0.008 | 27409.4 | 0.021 |
| MOPNR | 11360.1 | 0.011 | 17733.1 | 0.032 | 24871.2 | 0.064 |
| PPO-BiHyb | 8905.4 | 66.484 | 15192.2 | 149.215 | 22371.2 | 571.424 |
| Greedy (-) | 9300.4 | 0.219 | 16185.5 | 0.139 | 23788.9 | 0.143 |
| S(16) (-) | 9079.9 | 0.322 | 15974.1 | 0.340 | 23477.6 | 0.425 |
| S(64) (-) | 9037.3 | 0.467 | 15804.1 | 0.748 | 23412.2 | 1.233 |
| S(256) (-) | 8976.2 | 0.959 | 15684.9 | 2.338 | 23264.4 | 4.419 |
| Greedy (ours) | 8845.6 | 0.057 | 14981.2 | 0.100 | 22332.7 | 0.259 |
| S(16) (ours) | 8782.4 | 0.114 | 14972.0 | 0.287 | 22330.2 | 0.674 |
| S(64) (ours) | 8742.5 | 0.216 | 14968.1 | 0.699 | 22323.0 | 1.856 |
| S(256) (ours) | 8694.4 | 0.540 | **14964.7** | 2.270 | **22320.8** | 6.485 |

Table 11: Experiment results on 1000-node synthetic computation graphs with ablation studies. Greedy and Sampling methods with (-) indicate results from the model trained with greedy baseline and logit standardization in Gagrani et al. (2022).

|  | Layered Graph | | Erdos-Renyi | | Stoc. Block Model | |
|---|---|---|---|---|---|---|
|  | SpeedUp | Time (sec) | SpeedUp | Time (sec) | SpeedUp | Time (sec) |
| CP | 4.580 | 0.058 | 5.049 | 0.055 | 4.701 | 0.055 |
| SPT | 4.526 | 0.013 | 4.541 | 0.008 | 4.473 | 0.007 |
| MOPNR | 4.745 | 0.075 | 5.112 | 0.068 | 4.761 | 0.068 |
| Greedy (-) | 4.597 | 0.051 | 4.871 | 0.047 | 4.657 | 0.060 |
| S(16) (-) | 4.706 | 0.196 | 4.989 | 0.219 | 4.792 | 0.217 |
| S(64) (-) | 4.740 | 0.515 | 5.028 | 0.529 | 4.833 | 0.536 |
| S(256) (-) | 4.767 | 1.730 | 5.056 | 1.716 | 4.863 | 1.775 |
| Greedy (ours) | 4.819 | 0.094 | 5.194 | 0.046 | 4.866 | 0.043 |
| S(16) (ours) | 4.848 | 0.311 | 5.211 | 0.214 | 4.916 | 0.198 |
| S(64) (ours) | 4.872 | 0.750 | 5.227 | 0.542 | 4.944 | 0.477 |
| S(256) (ours) | **4.889** | 2.418 | **5.239** | 1.799 | **4.965** | 1.533 |

### G.3.2 REAL-WORLD GRAPHS

Our method outperforms Gagrani et al. (2022)'s method and shows comparable runtimes for all tasks (See Table 12.).

Table 12: Experiment results on real-world computation graphs with ablation studies. Greedy and Sampling methods with (-) indicate results from the model trained with greedy baseline and logit standardization in Gagrani et al. (2022).

| | 200 - 500 Node Graphs | | 500 - 700 Node Graphs | | 700 - 1000 Node Graphs | |
| --- | --- | --- | --- | --- | --- | --- |
| | SpeedUp | Time (sec) | SpeedUp | Time (sec) | SpeedUp | Time (sec) |
| Const. Prog. | 3.267 | - | 3.183 | - | 2.497 | - |
| CP | 3.174 | 0.007 | 2.804 | 0.016 | 2.739 | 0.025 |
| SPT | 3.107 | 0.002 | 2.868 | 0.005 | 2.664 | 0.008 |
| MOPNR | 3.181 | 0.009 | 2.825 | 0.020 | 2.739 | 0.028 |
| PPO-BiHyb | 3.223 | 17.937 | 2.965 | 52.777 | 2.798 | 322.793 |
| Greedy (-) | 3.227 | 0.196 | 3.114 | 0.028 | 2.822 | 0.045 |
| S(16) (-) | 3.248 | 0.243 | 3.151 | 0.153 | 2.843 | 0.238 |
| S(64) (-) | 3.254 | 0.324 | 3.179 | 0.404 | 2.847 | 0.656 |
| S(256) (-) | 3.258 | 0.644 | 3.192 | 1.317 | 2.853 | 2.205 |
| Greedy (ours) | 3.245 | 0.152 | 3.131 | 0.098 | 2.846 | 0.060 |
| S(16) (ours) | 3.271 | 0.192 | 3.188 | 0.245 | 2.848 | 0.230 |
| S(64) (ours) | 3.278 | 0.263 | 3.199 | 0.456 | 2.856 | 0.606 |
| S(256) (ours) | **3.286** | 0.595 | **3.207** | 1.309 | **2.860** | 2.001 |

# H  COMBINING SEARCH ALGORITHM WITH NEURAL SCHEDULER

We also conducted ablation studies to study the effect of using more powerful search algorithm with our method and analyze the performance-run time tradeoff against using sampled priorities. We combine our neural scheduler with genetic algorithm (GA) search. The initial population for GA is obtained by sampling 256 node priority vectors using our learned model. We let GA run for 100 generations with a population size of 1000. We use the publicly available GA implementaiton of Pymoo (Blank & Deb, 2020) for our experiments.

**Computation graph scheduling (synthetic).** We observe that for layered graphs and stochastic block model, using GA improves the performance marginally whereas there is no improvement for the Erdos-Renyi graphs. Also, the GA search increases the runtime significantly compared to the runtime of our method in sampling mode S(256).

Table 13: Experiment results on synthetic computation graphs with GA.

| | Layered Graph | | Erdos-Renyi | | Stoc. Block Model | |
| --- | --- | --- | --- | --- | --- | --- |
| | SpeedUp | Time (sec) | SpeedUp | Time (sec) | SpeedUp | Time (sec) |
| S(256) | 4.889 | 2.418 | 5.239 | 1.799 | 4.965 | 1.533 |
| S(256) + GA | 4.997 | 723.245 | 5.239 | 707.659 | 4.982 | 700.248 |

**Computation graph scheduling (real-world).** We found that using GA improves the speedup of schedules, while requiring significantly more computation time. The time required for GA increases as the graph size grows.

Table 14: Experiment results on real-world computation graphs with GA.

| | 200 - 500 Node Graphs | | 500 - 700 Node Graphs | | 700 - 1000 Node Graphs | |
| --- | --- | --- | --- | --- | --- | --- |
| | SpeedUp | Time (sec) | SpeedUp | Time (sec) | SpeedUp | Time (sec) |
| S(256) | 3.286 | 0.595 | 3.207 | 1.309 | 2.860 | 2.001 |
| S(256) + GA | 3.304 | 151.331 | 3.240 | 403.829 | 2.882 | 687.288 |

**DAG Scheduling (TPC dataset).** For TPC-H-50, we observed that GA search can improve the performance, whereas either no improvement (TPC-H-100) or negligible improvement (TPC-H-150) was observed for other datasets. Similar to computation graph scheduling tasks, significantly longer computation time is required, and the required time increases as the graph size increases.

Table 15: Experimental results on TPC-H datasets with GA.

| | TPC-H-50 | | TPC-H-100 | | TPC-H-150 | |
|---|---|---|---|---|---|---|
| | Makespan | Time (sec) | Makespan | Time (sec) | Makespan | Time (sec) |
| S(256) | 8694.4 | 0.540 | 14964.7 | 2.270 | 22320.8 | 6.485 |
| S(256) + GA | 8646.2 | 246.942 | 14964.7 | 965.119 | 22320.7 | 2089.547 |

