# OpenReview forum: "Neural DAG Scheduling via One-Shot Priority Sampling"
_ICLR.cc/2023/Conference — ICLR 2023 poster_

### Official Review · Reviewer_HjB3 · 2022-10-24

**Confidence:** 5
**Clarity, Quality, Novelty And Reproducibility:** The paper is well written and clear.
**Correctness:** 4
**Technical Novelty And Significance:** 2
**Empirical Novelty And Significance:** 2
**Recommendation:** 5

**Strength And Weaknesses:**

The novelty of the technique is mostly incremental - the proposed technique piggy back's on REINFORCE, Gumbel top-k trick and the list scheduling algorithm. The novelty is in how to make training with REINFORCE stable, which can be notoriously hard to get right.

What I found most interesting about the work was the authors' decision to create a Sampling version of the algorithm -- One important feature of the proposed technique is that the trained model can be used zero-shot once it has been trained on a dataset, making it extremely efficient compared to other neural-baselines. This is similar to the REGAL technique in Paliwal et al except that they use a genetic algorithm instead of the random search that the authors use in Sampling. Given that Sampling is always stronger (as it can't be worse than Greedy), I ask two things
- why didn't the authors consider using a more powerful search algorithm instead of just random search in Sampling?
- why didn't the authors consider using the cost of the of the best solution found by the search algorithm as the reward in REINFORCE instead of just using Greedy?


**Summary Of The Paper:**

The authors propose solving dag scheduling problems by using GNNs to provide node priorities in the list scheduling algorithm. The GNN is trained using good old REINFORCE. The training is made stable using logit renormalization and cost standardization, as introduced by the authors.

The authors introduce two versions of their Greedy and Sampling. In Sampling, they sample multiple node priorities from an encoded distribution. Greedy is a special case of Sampling, i.e. S(1).'

The authors evaluate their technique on several baselines, both real and synthetic. Their technique is consistently the best among all relevant baselines.

**Summary Of The Review:**

Overall, the paper is interesting, well written and well motivated. However, given that the novelty is incremental, the strength of the paper would lie in its empirical evaluation, hence I encourage the authors to explore more avenues such as how would their technique work using a more advanced search algorithm as proposed above.

---

> ### Author Response · Authors · 2022-11-16
> **[Reviewer HjB3, Q1] Difference between Greedy and S(1).**
>
> *(Comment from Reviewer HjB3: The authors introduce two versions of their Greedy and Sampling. In Sampling, they sample multiple node priorities from an encoded distribution. Greedy is a special case of Sampling, i.e. S(1).)*
>
> We would like to clarify that greedy is not the same as S(1). In the greedy inference mode, we generate the priorities without perturbing the logits, whereas we perturb the logits using Gumbel noise in the sampling inference mode. More precisely, node priorities for the greedy and sampling inference are obtained as follows:
> - Greedy
> $
> \vec{V}\_{greedy}=\mathrm{arg~sort}\_{v\in\mathcal{V}} \mathrm{logits}\_\theta(v;G)
> $
>
> - Sampling
> $\vec{V}\_{sampling}=\mathrm{arg~sort}\_{v\in\mathcal{V}} [\mathrm{logits}\_\theta(v;G)+Z(v) ]$.
>
> Here, $\mathcal{V}$ is a set of nodes, and $Z(v)$ are i.i.d. standard Gumbel random variables.

---

> ### Author Response · Authors · 2022-11-16
> **[Reviewer HjB3, Q2] Novelty concern.**
>
> *(Comment from Reviewer HjB3: The novelty of the technique is mostly incremental - the proposed technique piggy back's on REINFORCE, Gumbel top-k trick and the list scheduling algorithm. The novelty is in how to make training with REINFORCE stable, which can be notoriously hard to get right.)*
>
> We would like to kindly request Reviewer HjB3 to read **[General Response 1]** and **[General Response 2]**.

---

> ### Author Response · Authors · 2022-11-16
> **[Reviewer HjB3, Q3] Why did we consider random sampling instead of search algorithm?**
>
> *(Comment from Reviewer HjB3: Why didn't the authors consider using a more powerful search algorithm instead of just random search in Sampling?)*
>
> We agree with Reviewer HjB3 that adding a search algorithm can further improve the performance of our approach in terms of makespan. **However, search will incur additional computational complexity and increase the runtime of our algorithm.** One of the key focus areas of our research is ML compilers where good schedules need to be obtained within a very restricted amount of time. Although having better search can improve the performance, it will limit the usage of our algorithm in runtime-sensitive applications, e.g., computation graph scheduling on ML compilers. Also, we would like to highlight is that **our algorithm with random sampling consistently outperforms our ML baseline, PPO-BiHyb, where beam search was incorporated to find out better schedules.** This result supports the effectiveness of our approach in finding good schedules with significantly smaller runtime.

---

> ### Author Response · Authors · 2022-11-16
> **[Reviewer HjB3, Q4] Why didn’t we consider the cost of search algorithm’s best solution for REINFORCE?**
>
> *(Comment from Reviewer HjB3: Why didn't the authors consider using the cost of the of the best solution found by the search algorithm as the reward in REINFORCE instead of just using Greedy?)*
>
> As we discussed in **[Reviewer HjB3, Q3]**, one major contribution of our submission is achieving the best performance among heuristics while the minimum runtime is achieved among ML heuristics. To achieve this goal, we had to train our model without search so that we can expect shorter runtime during evaluation. Also, if we use the cost from search algorithms with randomness (e.g., genetic algorithm as Reviewer HjB3 recommended), the cost becomes stochastic even if the same policy output (node priorities) is used. This increases the variance of REINFORCE gradients and generally leads to longer training and unstable performance. On the other hand, we convert the priorities to a schedule by list scheduling which is a deterministic mapping and does not involve the aforementioned issue.

---

> ### Author Response · Authors · 2022-11-24
> **Gentle reminder for Reviewer HjB3**
>
> We would like to kindly ask Reviewer HjB3 if our response was sufficient to answer your questions and resolve your concerns. Since Discussion Stage 2 will last until December 12th (from Author Guide of ICLR 2023), we would like to hear your feedback at your earliest convenience and will be willing to resolve any remaining concerns. We truly appreciate your effort to review our work!

---

> > ### Comment · Reviewer_HjB3 · 2022-11-26
> > **Response**
> >
> > Thank you for responding to my queries.
> >
> > The authors raise a valid point about how more complicated search algorithms will increase the running time, making the technique unsuitable for compilers. I raise two counter points -
> >
> > 1. it is still an interesting research question that would be a valuable contribution for the domain.
> >
> > 2. ML models often have very large running times and one coud argue that if a slower compiler shaves off a few hours from training time, it might still be acceptable
> >
> >
> > About novelty:
> >
> > > However, if we restrict our problems of interest to scheduling, our work is the first one applying the Gumbel-top-k trick to a wide range of scheduling problems ...
> >
> > I agree with the authors evaluation that this is the first time I have seen the Gumbel top-k trick being applied to this domain. However, I retain my recommendation that this is not the strength not the main contribution of the paper.
> >
> > Many different techniques to solve graph scheduling problems have been proposed in literature, especially in a fast ML compiler setting, each evaluating their performance on their own set of relevant baselines.
> >
> > The authors provide good evidence that this technique is good and might be practical for the compiler setting. However, due to how fragmented this domain is, it would just be another ML based optimizer to solve graph scheduling problems.
> >
> > This is why I feel the strength of the paper should have been a robust empirical evaluation of the technique, stretching it to its limit.
> >
> > Overall, I find this paper sort of interesting but not a breakthrough. I will retain my recommendation score.

---

> > > ### Author Response · Authors · 2022-11-30
> > > **2nd response to Reviewer HjB3 (1/2)**
> > >
> > > We appreciate for your valuable comments to our author response. We summarized our additional response below, point-wisely. We hope that our 2nd response is sufficient to address Reviewer HjB3's concerns.
> > >
> > > *(Comment: The authors raise a valid point about how more complicated search algorithms will increase the running time, making the technique unsuitable for compilers. I raise two counter points - 1. it is still an interesting research question that would be a valuable contribution for the domain.)*
> > >
> > > We agree that using search is an interesting topic by itself. However, combining search is quite orthogonal to our work's motivation; minimizing the runtime of algorithm when deployed while outperforming our baselines. As we described in our initial author response, we empirically showed that our algorithm already strictly outperforms our baselines without using search algorithms, which is sufficient to support the empirical significance and motivation of our work. One can easily expect that using search as a post-processing step will result in the performance at least as good as the generated solution from our algorithm but with increased runtime. For this reason, we believe not combining our algorithm with search cannot weaken the contribution of our work.
> > >
> > > To empirically support our claim and see if our algorithm can be easily combined with search, **we did additional experiments with genetic algorithm (GA) as suggested by Reviewer HjB3’s initial review**. We use Pymoo’s GA implementation that provides the publicly available GA implementation. We sample 256 node priorities by using our learned model to acquire initial population, and GA is applied on top of those sampled node priorities. We use 100 generations with 1000 population size. Due to the restricted amount for response, we evaluated our model in almost all tasks except JSSP, but we expect the tendency will be the same for JSSP.
> > >
> > > **Computation graph scheduling (synthetic).** For Layered Graph and Stochastic Block Model, using GA improves performance with a huge increase in runtime, whereas there is no improvement for Erdos-Renyi.
> > > |                                | Layered Graph |            | Erdos-Renyi |            | Stock. Block Model |            |
> > > |--------------------------------|---------------|------------|-------------|------------|--------------------|------------|
> > > |                                | SpeedUp       | Time (sec) | SpeedUp     | Time (sec) | SpeedUp            | Time (sec) |
> > > | Sampling (256)            | 4.889         | 2.418      | **5.239**       | 1.799      | 4.965              | 1.533      |
> > > | Sampling (256)  + GA  | **4.997** | 723.245 | **5.239**       | 707.659      | **4.982**     | 700.248      |
> > >
> > > **Computation graph scheduling (real-world).** We found that using GA improves the speedup of schedules but requires significantly more computation time. The time required for GA increases as the graph size increases.
> > > |                                | 200-500 Node Graphs |            | 500-700 Node Graphs |            | 700-1000 Node Graphs |            |
> > > |--------------------------------|---------------------|------------|---------------------|------------|----------------------|------------|
> > > |                                | SpeedUp             | Time (sec) | SpeedUp             | Time (sec) | SpeedUp              | Time (sec) |
> > > | Sampling (256)  | 3.286      | 0.595      | 3.207               | 1.309      | 2.86                 | 2.001      |
> > > | Sampling (256)  + GA   | **3.304**     | 151.331      | **3.240**     | 403.829      | **2.882**       | 687.288      |
> > >
> > > **DAG Scheduling (TPC dataset).** For TPC-H-50, we could find out better results, whereas either no improvement (TPC-H-100) or negligible improvement (TPC-H-150) was observed from other datasets. Similar to computation graph scheduling tasks, significantly longer computation time is required, and the required time increases as the graph size increases.
> > > |                                | TPC-H-50 |            | TPC-H-100 |            | TPC-H-150 |            |
> > > |--------------------------------|----------|------------|-----------|------------|-----------|------------|
> > > |                                | Makespan | Time (sec) | Makespan  | Time (sec) | Makespan  | Time (sec) |
> > > | Sampling (256)  | 8694.4   | 0.54       | **14964.7**   | 2.27       | 22320.8   | 6.485      |
> > > | Sampling (256)  + GA    | **8646.2**   | 246.942       | **14964.7**   | 965.119       | **22320.7**   | 2089.547      |

---

> > > ### Author Response · Authors · 2022-11-30
> > > **2nd response to Reviewer HjB3 (2/2)**
> > >
> > > *(Comment: - 2. ML models often have very large running times and one could argue that if a slower compiler shaves off a few hours from training time, it might still be acceptable.)*
> > >
> > > From our best understanding, what Reviewer HjB3 considered is the training pipeline of a ML model, and a compiler is used to optimize the forward and backward computation graphs of that ML model. Assuming that our understanding is correct, we agree that using search may be beneficial. For example, we can let our algorithm quickly generate solutions and use search algorithm to improve those solutions (e.g., genetic algorithm with populated solutions), where our algorithm works as a fast initializer with good performance.
> > >
> > > However, we would like to argue that the above example is not time-sensitive. **As a time-sensitive example, one can consider quickly running ML models for inference and suppose that models to run are varying on the fly (e.g., due to varying target applications)**. In this case, we need to quickly compile each ML graph instance to achieve better inference speed. Similarly, there are **various practical cases where real-time DAG inputs are given and we have to schedule those as quickly as possible**, e.g., cluster resource management. (Note that we consider general scheduling problems which are not confined into ML computation graphs.)
> > >
> > > *(Comment: About novelty: "However, if we restrict our problems of interest to scheduling, our work is the first one applying the Gumbel-top-k trick to a wide range of scheduling problems ..." I agree with the authors evaluation that this is the first time I have seen the Gumbel top-k trick being applied to this domain. However, I retain my recommendation that this is not the strength not the main contribution of the paper.)*
> > >
> > > We mentioned Gumbel-top-k trick in our **[General Response 1]** since another reviewer (Reviewer p2nE) mentioned using Gumbel-top-k is not new from the initial review. We would like to clarify that **the contribution of our work is on using one-shot learning-based node priorities together with list scheduling**, where Gumbel-top-k is used to model correct probability distributions over priorities and sample node priorities in parallel. As far as we know, any similar approach has not been proposed so far.
> > >
> > > *(Comment: Many different techniques to solve graph scheduling problems have been proposed in literature, especially in a fast ML compiler setting, each evaluating their performance on their own set of relevant baselines.)*
> > >
> > > As far as we know, **we included almost all strong ML scheduling baselines** (except REGAL [18] that does not have a publicly available code). We found that none of those baselines considered reducing runtime and improving computational efficiency, whereas we fairly considered runtime together with strictly better performance.
> > >
> > > *(Comment: The authors provide good evidence that this technique is good and might be practical for the compiler setting. However, due to how fragmented this domain is, it would just be another ML based optimizer to solve graph scheduling problems.)*
> > >
> > > Together with JSSP and DAG scheduling, ML compiler is one of the scheduling domains we evaluated. We agree that the field of ML schedulers is fragmented, which is why we evaluated our method across multiple scheduling domains in our work. We would like to emphasize that **our baselines were focused on only one of the domains in our work (one of DAG scheduling/JSSP/ML computation graphs), which we believe supports the strength of our work**.
> > >
> > > *(Comment: This is why I feel the strength of the paper should have been a robust empirical evaluation of the technique, stretching it to its limit.)*
> > >
> > > If "stretching it to its limit" means using search with our algorithm, **not using search with our algorithm cannot weaken our contribution since our algorithm is already strictly outperforming our baselines without search**, and it is clear that search will only improve the performance (with longer runtime) as shown in our additional experiments.

---

### Official Review · Reviewer_zenr · 2022-10-24

**Confidence:** 3
**Clarity, Quality, Novelty And Reproducibility:** I do not see any problems with clarit…
**Correctness:** 4
**Technical Novelty And Significance:** 4
**Empirical Novelty And Significance:** 4
**Recommendation:** 8

**Strength And Weaknesses:**

Strength:
* The problem being addressed by the proposed method is clearly motivated and explained. It also has a big impact, as any scheduling approach must have low runtime to be usable in practical scenarios. The approach considered within this work is novel to the best of my knowledge (i.e., the differences from Gagrani et. al. seem to be significant/non-trivial, as you must find a way to obtain the schedule in a single shot).
* The use of Gumbel variables and the Gumbel top-k trick is explained well and interesting.
* The method seems to perform well empirically relative to neural and heuristic baselines.

Weakness:
* It would be good to provide discussion of how long it takes to train your method, how much data is needed, etc. This is probably an additional latency consideration that would be of interest to anyone trying to practically use your method. I think this discussion is needed to more thoroughly compare your approach to common, heuristic methods on a basis of practicality.
* Time is still worse compared to heuristic baselines, but it is still close. It seems like to achieve the best performance, more time is needed. However, this is somewhat unavoidable for approaches that leverage neural networks.
* Outperformed by constraint programming on TPC-H, but this doesn’t seem to be a big issue. It is only in one case and performance of the proposed method is still pretty similar.


Small Comment:
* Logit norm regularization explanation is well done. I like how you motivate the problem that exists using binary variables, then propose a solution that works well empirically. This explanation makes sense and is non-trivial.
* In the summary of contributions “(w.r.t to the makespan)”, you can remove the “to”.

Questions:
* Is REINFORCE the only way to train this algorithm? Are there potentially other methods (maybe even non-RL) that would be less data hungry or provide a better result? It would be good to understand a little better why REINFORCE is chosen.

**Summary Of The Paper:**

This paper considers node/operation scheduling within DAGs, which is previously solved using heuristics but more recently approached with ML-based solutions. ML techniques are typically more costly, but the authors employ a “one-shot” encoder networks to sample node priorities in a single forward pass of the network. These priorities are then converted via list scheduling to the final schedules. This encoder can sample priorities in parallel, which makes it much faster than baselines. Although the method is still a bit slower than heuristic techniques in most cases, the proposed method is shown to actually outperform both heuristic and ML-based techniques.


**Summary Of The Review:**

Personally, I am not familiar with reinforcement learning. As such, I am completely open to discussion of my comments with other authors/reviewers. This review reflects my initial impression of the paper given that I am somewhat unfamiliar with this area (DAG scheduling/RL). I emphasize my final score will be mostly based upon subsequent discussion that I hope will allow me to gain more perspectives on the work.

I think the paper is well-written, motivated nicely, and novel (to the best of my knowledge). Empirically, the method performs very well (best in almost every case). As such, I feel that this paper provides a nice contribution to the community that is novel (AFAIK), performant, and practical. In its current form, my biggest recommendation to the authors would be to provide more details on training complexity and requirements. To better understand how using your method compares to using heuristic baseline methods, this discussion is needed. Yes, the method performs very well. But, practitioners may not use it if too much effort is required. Currently, I lack from the paper an idea of where your method stands on this spectrum -- is it practical enough to actually use?

Nonetheless, I thank the authors for the nicely-written and interesting paper. I think the paper is a good contribution, and I look forward to future discussion!

---

> ### Author Response · Authors · 2022-11-16
> **[Reviewer zenr, Q1] Training time and the number of training data.**
>
> *(Comment from Reviewer zenr: It would be good to provide discussion of how long it takes to train your method, how much data is needed, etc. This is probably an additional latency consideration that would be of interest to anyone trying to practically use your method. I think this discussion is needed to more thoroughly compare your approach to common, heuristic methods on a basis of practicality.)*
>
> We thank the reviewer for the thoughtful question. In general, we found training with our method to be quick taking from a few minutes to a few hours to converge for the tasks that we considered. For readers who would like to see the practicality of our algorithm, we provide how long it takes to roughly converge and how many epochs are required to converge for all tasks in below and Appendix F.1.
> | Experiment                                      |  Convergence time |  Time per epoch |  Num. epochs for convergence  |
> |-------------------------------------------------|-------------------|-----------------|-------------------------------|
> | Real-world computation graphs                   | 25 mins           | 3 mins          | 8                             |
> | Synthetic graphs - Layered (500)                | 2.5 hours         | 1.2 hours       | 2                             |
> | Synthetic graphs - Erdos-Renyi (500)            | 3 hours           | 1.2 hours       | 2.5                           |
> | Synthetic graphs - Stochastic Block Model (500) | 3.5 hours         | 1.2 hours       | 3                             |
> | Synthetic graphs - Layered (1k)                 | 3 hours           | 2.4 hours       | 1.25                          |
> | Synthetic graphs - Erdos-Renyi (1k)             | 8 hours           | 2.4 hours       | 3.3                           |
> | Synthetic graphs - Stochastic Block Model (1k)  | 8 hours           | 2.4 hours       | 3.3                           |
> |                                                                    |                          |                       |                                 |
> | TPC-H dataset - TPC-50                          | 1 hour            | 3.6 min         | 16.7                          |
> | TPC-H dataset - TPC-100                         | 5 hours           | 14.4 min        | 20.8                          |
> | TPC-H dataset - TPC- 150                        | 16 hours          | 28.8 min        | 33                            |
> |                                                                    |                          |                       |                                 |
> | JSSP dataset - (25, 10)                          | 30 mins           | 2 min           | 15                            |
> | JSSP dataset - (25, 20)                          | 45 mins           | 4.5 min         | 10                            |
> | JSSP dataset - (25, 30)                          | 1 hour            | 7.5 min         | 8                             |
> | JSSP dataset - (50, 10)                          | 45 mins           | 4.5 min         | 10                            |
> | JSSP dataset - (50, 20)                          | 3 hours           | 12 min          | 15                            |
> |                                                                    |                          |                       |                                 |
>
> Overall, we observe that as the graph size increases, required training time for each epoch becomes longer. Although this is inevitable due to the increasing computation time (for both neural network and list scheduling) as the graph size becomes larger, we observe that our algorithm trains much faster in general compared to our ML baseline due to the one-shot nature of our algorithm. Regarding the number of training data, we would like to note that provided this information in our submission (Appendix C and F). We list them as follows for Reviewer’s convenience; the numbers of training graphs are 100, 50, 3000, and 92 for tasks with JSSP, TPC-H, synthetic and real-world computation graphs, respectively. Note that our algorithm works well for various number of training graphs, while outperforming our baselines.
>
> To address the comment on practicality, we believe training with our method converges in a fast manner which is evident from the above training times. The single-shot nature of our scheme also plays a role in achieving low training time.  Also, our method does not necessarily require a large training set to generalize well. This is evident from the experiments on the TPC dataset where the training set has only 50 graphs. Hence, overall we have strong conviction that our method can be trained easily without requiring a large dataset and will be useful to practitioners solving scheduling problems.
> We hope this information benefits the readers who are interested in our work from practical perspectives.

---

> > ### Comment · Reviewer_zenr · 2022-11-21
> > **Response to authors**
> >
> > Thank you so much to the authors for their thoughtful and careful responses. I believe most of my questions with this work are answered. After gaining a more careful understanding, it seems that the proposed technique is novel (i.e., outperforms previous baselines consistently due to proposed methodological changes), practical (i.e., training/evaluation times and amount of data needed are reasonable), and work really well compared to baselines if you consider the solver's time and quality of solution. With this in mind, I increase my score to reflect my improved view of the paper.
> >
> > I am happy to further discuss concerns with authors/other reviewers as we continue to progress towards a decision. My current opinion is that the paper is sound and meets requirements for acceptance.

---

> > > ### Author Response · Authors · 2022-11-24
> > > **Comment to Reviewer zenr**
> > >
> > > We are glad to see that our responses answer your questions and resolve your concerns. We truly appreciate your increasing the score of our work!

---

> ### Author Response · Authors · 2022-11-16
> **[Reviewer zenr, Q2] Runtime is close to but worse than non-ML baselines.**
>
> *(Comment from Reviewer zenr: Time is still worse compared to heuristic baselines, but it is still close. It seems like to achieve the best performance, more time is needed. However, this is somewhat unavoidable for approaches that leverage neural networks.)*
>
> The heuristic methods used in our submission derive the node priorities from cheaply computable node characteristics, which makes those methods run very fast compared to ML approaches. We believe it is almost impossible for ML approaches to beat non-ML heuristics in this case since even a single forward computation of neural networks usually requires longer computation time. However, we empirically show that our greedy-mode algorithm runs comparably with those heuristics, while our algorithm always requires much shorter runtime than our ML baselines, which supports the effectiveness of our one-shot scheduler.

---

> ### Author Response · Authors · 2022-11-16
> **[Reviewer zenr, Q3] Our algorithm is outperformed by Constraint Programming Solver in TPC-H-50.**
>
> *(Comment from Reviewer zenr: The proposed algorithm is outperformed by constraint programming on TPC-H, but this doesn’t seem to be a big issue. It is only in one case and performance of the proposed method is still pretty similar.)*
>
> Although the constraint programming solver outputs the exact solution in TPC-H-50, it should be noted that we allow the time limit of 24 hours for the solver. (We described this in the appendix of our submission.) Among 10 test graphs of TPC-H-50, we could get the optimal solutions for 3 graphs, where it took 2,429 seconds, 21,949 seconds and 39,274 seconds to acquire those optimal solutions. For the other 7 test graphs, the solver outputs the best solution that it could find out within 24 hours. Therefore, the solver requires significantly longer time than our algorithm to get solutions, which makes it far from being practical.
>
> While our ML baseline, PPO-BiHyb [16], did not consider the solver’s performance in TPC-H dataset, we experimented the solver to confirm (1) the practicality of ML and non-ML heuristic algorithms in the dataset and (2) how much good performance those heuristics can achieve relative to the true optimum if available. In TPC-H-50 (where the average number of nodes is 467.2), **our algorithm performs the best among all ML and non-ML heuristics**, and its average makespan turns out to be closest to the solver’s makespan. As the graph size becomes larger (TPC-H-100, TPC-H-150, where the average numbers of nodes are 929.8 and 1384.5, respectively), we observed that the solver does not scale; the solver is even outperformed by non-ML heuristics in TPC-H-100 and encounters the out-of-memory issue in TPC-H-150. Therefore, we would like to claim that the empirical result in TPC-H dataset supports the scalability of our algorithm to larger graphs as well as the superiority of our algorithm among all heuristics.

---

> ### Author Response · Authors · 2022-11-16
> **[Reviewer zenr, Q4] Comment on logit norm regularization.**
>
> *(Comment from Reviewer zenr: Logit norm regularization explanation is well done. I like how you motivate the problem that exists using binary variables, then propose a solution that works well empirically. This explanation makes sense and is non-trivial.)*
>
> We appreciate Reviewer zenr’s comment on our explanation on a binary random variable example. We would like to link that we did some ablation studies to compare our algorithm with [1]’s algorithm in **[General Response 2]**, which is relevant to logit norm regularization. We wish our additional experiments strengthen your opinion.

---

> ### Author Response · Authors · 2022-11-16
> **[Reviewer zenr, Q5] Typo.**
>
> *(Comment from Reviewer zenr: In the summary of contributions “(w.r.t to the makespan)”, you can remove the “to”.)*
>
> We reflected this into our revision.

---

> ### Author Response · Authors · 2022-11-16
> **[Reviewer zenr, Q6] Why should we use REINFORCE?**
>
> *(Comment from Reviewer zenr: Is REINFORCE the only way to train this algorithm? Are there potentially other methods (maybe even non-RL) that would be less data hungry or provide a better result? It would be good to understand a little better why REINFORCE is chosen.)*
>
> Using RL may not be necessary. For example, using supervised learning w.r.t. optimal solver’s solutions is possible. However, acquiring those solutions may not be feasible if the problem size becomes large due to the combinatorial nature of our scheduling problems (which are NP-hard problems). Due to this reason, researchers have typically resorted to RL which does not require labelled dataset for training. REINFORCE is a well-known policy gradient algorithm in RL literature. Despite its simplicity, the so-called vanilla REINFORCE is avoided due to its inherent high gradient variance. Thankfully, there have been practical techniques (such as PPO [13] and GAE [14]) to reduce the variance in RL, so we adopted those in our submission. REINFORCE is also useful when the learning objective (reward) is non-differentiable or black-box (i.e., we do not know whether the objective is differentiable or not), which is appropriate for our setting. Such non-differentiability is relevant to **[Reviewer p2nE, Q3]** where Reviewer p2nE proposed reparameterization to reduce the variance of policy gradient.

---

### Official Review · Reviewer_p2nE · 2022-10-25

**Confidence:** 4
**Correctness:** 3
**Technical Novelty And Significance:** 2
**Empirical Novelty And Significance:** 3
**Recommendation:** 5

**Clarity, Quality, Novelty And Reproducibility:**

The paper is presented clearly, and the quality of the current execution is ok but some more experiments can be included. The reproducibility seems fine as the approach is relatively simple. The main concern is on the novelty and the technical quality.


**Strength And Weaknesses:**

Strength:
- The paper is well written and the background and problem is described clearly.
- The empirical results seem to be good.

Weakness:
- Technically the paper is relatively straightforward and lacks novelty in terms of the technical contribution. Leveraging policy gradient or RL in general for combinatorial optimization has been studied over the past several years. The main contribution of the paper is to use a variant of the parameterization of the policy, where the encoder is from another recent paper and the gumbel-topk used for decoding from the encoded logits is not new.
- Continuing with above, the technical part can be potentially improved in following aspects:
1) It seems that the current approach doesn’t guarantee that the generated permutation satisfies the topological order specified by the DAD (correct me if I’m wrong). It would be nice to make it more compatible with the problem that is being solved, instead of using a generic approach in a straightforward way.
2) The nice thing about the gumbel trick is that it admits a reparameterization. If the author can build something that allow end to end optimization with gradient propagates from the objective to the graph encoder, then it would be nicer and more efficient than REINFORCE, hopefully.

- The paper claims that the one-shot decoding is beneficial. However it seems that it is a surrogate of the model used in [1], as mentioned by the authors in Sec 3.2. So a natural question would be, how does [1] perform in the experiments presented in the paper? The hypothesis is that [1] would yield better results but worse speed, so a trade-off is expected. More comprehensive comparison with [1] would be needed for the experimental sections.

- The application for the scheduling seems to be specific, but the technique using GNN + gumbel-topk with RL seems to be general. To make it naturally aligned with the technique, there could be several ways to improve:
1) include other combinatorial optimizations that optimize for a permutation, like the TSP mentioned by the authors;
2) make the approach more appropriate and adapted to the DAG based optimization, as I mentioned above about the techniques.

References:
[1] Neural topological ordering for computation graphs, Gagrani et.al


**Summary Of The Paper:**

This paper proposes to use policy gradients to learn the permutation of node sequences for applications in scheduling operations. Specifically the model takes a pre-defined DAG structure which describes the job dependencies as input, and produces the logits for each job. The permutation is sampling via gumbel-topk trick and then REINFORCE gradient is used to update the policy network. Experiments on some synthetic and real-world scheduling tasks show some improvement over existing approaches.


**Summary Of The Review:**

Overall this paper presents a simple yet efficient approach for the scheduling problem. There are several aspects that need to be justified, in terms of the 1) experimental quality and thoroughness; 2) technical correctness/improvements; 3) generality to more CO problems that optimize for a permutation.

---

> ### Author Response · Authors · 2022-11-16
> **[Reviewer p2nE, Q1] Our algorithm seems lack of novelty.**
>
> *(Comment from Reviewer p2nE: Technically the paper is relatively straightforward and lacks novelty in terms of the technical contribution. Leveraging policy gradient or RL in general for combinatorial optimization has been studied over the past several years. The main contribution of the paper is to use a variant of the parameterization of the policy, where the encoder is from another recent paper and the Gumbel-top-k used for decoding from the encoded logits is not new.)*
>
> We would like to kindly refer Reviewer p2nE to check **[General Response 1]**.

---

> > ### Comment · Reviewer_p2nE · 2022-11-26
> > **Re: Q1**
> >
> > Thanks for your further explanations. I agree that conditioning on this specific problem, the paper proposed a new composition of existing techniques. The main reason why I find it didn't meet the standard at ICLR is that, the combination of techniques is done in a relatively straightforward way, and in my opinion it can be applied to other problems with minor modification. We probably won't consider that "directly applying this technique combo once again on a new problem" to be novel, as otherwise many more papers could be generated in this way.
> >
> > That's why I explained in my review what might be better in terms of the technical contribution: 1) demonstrate that the technique is general for a family of problems; 2) adapt the techniques in a more principled way for the specific problem, for example how to do one-shot sampling that guarantees the ordering constraints without post-processing.

---

> > > ### Author Response · Authors · 2022-11-30
> > > **2nd response to Reviewer p2nE**
> > >
> > > We appreciate Reviewer p2nE’s feedback on our responses. We made further responses below. We hope that our 2nd response is sufficient to address the concerns.
> > >
> > > *(Comment: Thanks for your further explanations. I agree that conditioning on this specific problem, the paper proposed a new composition of existing techniques. The main reason why I find it didn't meet the standard at ICLR is that, the combination of techniques is done in a relatively straightforward way, and in my opinion it can be applied to other problems with minor modification. We probably won't consider that "directly applying this technique combo once again on a new problem" to be novel, as otherwise many more papers could be generated in this way.)*
> > >
> > > We would like to emphasize that our work is not a straightforward combination for a scheduling problem. We believe this is clearly proved by our new experiment from our initial author response.
> > > - Gagrani et al. [1] used Topoformer for sequencing problem. We used Topoformer for scheduling problem with list scheduling. We believe the novelty concern was raised due to using existing techniques (Topoformer, REINFORCE, Gumbel-Top-k), but** the fact that none of ML schedulers has tried to learn node priorities for list scheduling makes our method novel**. Together with this fact, we show that **our method strictly outperforms our baselines, while achieving much shorter runtime**.
> > > - We also pointed out **the issue of Topoformer’s logit standardization that restricts representation power of the model, which has not been discovered and supports the non-straightforward contribution of our method**. We proved this by using a simple example (binary random variable example) and a variety of experiments (the ablation study in our author response). This shows that our technical contributions (logit norm regularization and return-normalized REINFORCE) are necessary to achieve outstanding performance and supports the technical novelty of our work.
> > >
> > > *(Comment: That's why I explained in my review what might be better in terms of the technical contribution: 1) demonstrate that the technique is general for a family of problems.)*
> > >
> > > Again, **we consider a wide range of scheduling problems, whereas each has been separately considered in earlier works**. As far as we know, **our work is the first work across multiple scheduling problem instances including JSSP, DAG Scheduling, ML computation graphs, where each of them is working on DAGs**. We believe not pursuing generalization cannot weaken our work’s contribution since the current version is already general.
> > >
> > > *(Comment: 2) adapt the techniques in a more principled way for the specific problem, for example how to do one-shot sampling that guarantees the ordering constraints without post-processing.)*
> > >
> > > List scheduling is a widely adopted principled way of converting priorities (that do not have to satisfy the precedence constraints) to valid schedules (satisfying the precedence constraints) in operations research. Since list scheduling can be efficiently implemented, our ML model only needs to focus on how to sample node priorities without considering precedent constraints. This gives a strong advantage to the ML model since priorities can be sampled in parallel thanks to the Gumbel-top-k trick. We believe post-processing for DAGs is a necessary step to make solutions satisfying precedence constraints and our current way of strengthening ML model’s parallelism using Gumbel-top-k trick is a principled approach.

---

> ### Author Response · Authors · 2022-11-16
> **[Reviewer p2nE, Q2] Priority does not satisfy topological ordering. Training pipeline with list scheduling seems straightforward.**
>
> *(Comment from Reviewer p2nE:  It seems that the current approach doesn’t guarantee that the generated permutation satisfies the topological order specified by the DAG (correct me if I’m wrong). It would be nice to make it more compatible with the problem that is being solved, instead of using a generic approach in a straightforward way.)*
>
> First, we want to emphasize that the schedule generated by our algorithm satisfies the precedence constraints specified by the DAG.  If the reviewer is referring to the priority vector $\vec{V}$ by the *generated permutation* in his/her statement, then we would like to clarify that priority vector $\vec{V}$ does not need to satisfy topological order. The priority vector $\vec{V}$ generated by our algorithm is used as an input for the list scheduling that generates the final schedule which satisfies the precedence constraints specified by the DAG. For example, let us consider a chain graph of 3 nodes ($1 \rightarrow 2 \rightarrow 3$). Consider a priority vector (list of nodes, nodes on left have higher priority) $\vec{V}\_0 = [3, 2, 1]$, where $\vec{V}\_0$ does *not* satisfy topological ordering. As we described in **(Step 2)** in Section 2.2. of our submission, list scheduling considers the set of *ready nodes* (and ignores the rest of nodes not in the set) at each time and schedules the node with the highest priority in set of *ready nodes*. Note that there always should be only 1 ready node at each time for a chain graph. Therefore, list scheduling with the priority $\vec{V}\_0$ will result in a valid schedule $1 \rightarrow 2 \rightarrow 3$, i.e., each node is executed after its predecessor is finished.
>
> If reviewer’s statement *using a generic approach in a straightforward way* refers to using list scheduling in our pipeline, then we would like to respectfully disagree with reviewer’s statement. Our approach is simple, but it is not straightforward. Finding good schedules with list scheduling requires the right priority vector, otherwise it can generate sub-par schedules due to its greedy nature. While traditional methods have relied on using simple heuristics to generate node priorities, our approach generates the node priorities using a machine learning model. In fact, many prior works [2, 16, 17] have considered machine learning aiding of classical algorithms, and our method is also an instance of this class of research. Our experiments show that our model outperforms the hand-designed heuristics in terms of makespan as well as our ML baselines (which have more complex structure due to their multi-round processing) in terms of both makespan and runtime.

---

> > ### Comment · Reviewer_p2nE · 2022-11-26
> > **RE: Q2**
> >
> > thanks for your reply! I thought one would only rely on the priority vector for the decoding. In this case the proposed method also has the post-processing that guarantees the constraint satisfaction. If I understand correctly, any priority vector would be fine as the post-processing would make the final guarantee.
> >
> > I though it would be more interesting if a decoded priority vector itself is a valid ordering. This would be technically nontrivial.

---

> > > ### Author Response · Authors · 2022-11-30
> > > **Comment to Reviewer p2nE**
> > >
> > > We appreciate Reviewer p2nE's comment. We answered this comment in **2nd response to Reviewer p2nE** below.

---

> ### Author Response · Authors · 2022-11-16
> **[Reviewer p2nE, Q3] Why not use Gumbel reparameterization instead of REINFORCE?**
>
> *(Comment from Reviewer p2nE: The nice thing about the Gumbel trick is that it admits a reparameterization. If the author can build something that allow end to end optimization with gradient propagates from the objective to the graph encoder, then it would be nicer and more efficient than REINFORCE, hopefully.)*
>
> We appreciate Reviewer p2nE’s comment. We also thought that Gumbel reparameterization [11, 12] can reduce the variance of stochastic gradients, but there are a few reasons why we did not consider reparameterization in our submission.
> - **End-to-end optimization with Gumbel reparameterization requires cost function approximation.**  We recall that our learning objective (in Eq. (9)) is
> $
> \mathrm{arg~min}\_{\theta} \mathbb{E}\_{G\sim\mathcal{G}}\mathbb{E}\_{\vec{V}\sim\pi\_\theta(\cdot|G)} C(\vec{V};G),
> $
> where $\mathcal{G}$ is the training graphs, $\vec{V}$ is the sequence of nodes (list) by following node priorities, and $C(\vec{V};G)$ is the cost (makespan of the schedule from list scheduling). Since our cost is determined by list scheduling, *it is generally not differentiable*, which requires us to build a cost function approximator. **This will complicate training process (due to intermediate cost approximation), require more memory, increase the training time, and accumulate gradient error caused by the cost estimation.**
> -  **Our algorithm with REINFORCE is sufficiently simple, stable and efficient while outperforming our baselines.** As we mentioned in our submission, REINFORCE (or policy optimization) with cost (or advantage) standardization is already a widely adopted technique to stabilize training, e.g., in PPO [13] and GAE [14] implementations, and we used that.

---

> > ### Comment · Reviewer_p2nE · 2022-11-26
> > **RE: Q3**
> >
> > thanks for your reply! I was trying to help on the methodology side with something that is not that widely used. Maybe it can potentially lead to something more interesting/suitable for your specific problem. It is totally fine with the current approach.

---

> > > ### Author Response · Authors · 2022-11-30
> > > **Comment to Reviewer p2nE**
> > >
> > > We're happy to see that this concern is resolved. We appreciate your response.

---

> ### Author Response · Authors · 2022-11-16
> **[Reviewer p2nE, Q4] More comprehensive comparison with [1] is needed.**
>
> *(Comment from Reviewer p2nE: The paper claims that the one-shot decoding is beneficial. However, it seems that it is a surrogate of the model used in [1], as mentioned by the authors in Sec 3.2., so a natural question would be, how does [1] perform in the experiments presented in the paper? The hypothesis is that [1] would yield better results but worse speed, so a trade-off is expected. More comprehensive comparison with [1] would be needed for the experimental sections.)*
>
> We would like to kindly request Reviewer p2nE to read **[General Response 2]**.

---

> ### Author Response · Authors · 2022-11-16
> **[Reviewer p2nE, Q5] Algorithmic improvement via task generalization & DAG-based optimization**
>
> *(Comment from Reviewer p2nE: The application for the scheduling seems to be specific, but the technique using GNN + Gumbel-top-k with RL seems to be general. To make it naturally aligned with the technique, there could be several ways to improve; 1) include other combinatorial optimizations that optimize for a permutation, like the TSP mentioned by the authors; 2) make the approach more appropriate and adapted to the DAG based optimization, as I mentioned above about the techniques.)*
>
> ### **Algorithmic improvement via task generalization.**
> We believe our idea (one-shot priority sampling via Gumbel-top-k) can be well-extended to other CO problems by considering the following items:
> - *(Item 1) What is a proper graph encoder?* We choose Topoformer encoder by [1] since it is designed to efficiently abstract an input DAG’s information. However, this encoder may not be appropriate if other CO problems are considered.
> - *(Item 2) What is a proper mapping from priorities to problem-specific outputs?* We mapped each priority to a valid schedule by using list scheduling, but this should be modified if the goal problem is changed.
>
> As an example, we would like to provide brief ideas for those items when **Euclidean TSP** is the problem of our interest.
> - For (Item 1), we no longer can use Topoformer encoder since we consider fully connected input. Assuming that the input features are each node’s Euclidean coordinates, we think any permutation-equivariant encoder can be used, e.g., standard Transformer without positional encoding, set Transformer, standard GAT with fully connected message passing, etc. Note that the permutation equivariance property is required since we need acquire permuted priorities (across nodes) corresponding to permuted input node features.
> - For (Item 2), we can define the following mapping from priorities to form complete tours in TSPs; we argsort nodes by priorities and connect each node by following the order of nodes; we then connect the first and last nodes so that a complete tour is constructed. For each complete tour, we can evaluate the tour length and use it as a cost value for policy gradient.
>
> Since the above generalization experiments require too many modifications upon the code we already have, we didn’t pursue the experiments during Phase 1. Also, our submission is focused on devising ML scheduler that generally works well on a wide range of scheduling problems. In that sense, **we would like to emphasize that our algorithm is sufficiently general in scheduling tasks.** In our submission, the problems of our interest includes (1) Jop Shop Scheduling Problem (JSSP), (2) DAG scheduling, and (3) computational graph scheduling. It should be noted that our ML baselines only focus on one of those problems:
> - [19] proposed Learning-to-Dispatch (L2D) that is designed specifically for JSSP and **cannot be generalized to general DAGs due to JSSP’s chain-graph nature.**
> - [16] proposed PPO-BiHyb that works with DAG scheduling where **only scheduling for homogeneous machines is considered**.
> - [18] proposed REGAL that only aims to schedule **computational graphs**. (We didn’t compare our algorithm with this baseline since the authors did not make their code publicly available.)
>
> As far as we know, we present **the first algorithm that solves all those scheduling algorithms** and significantly outperforms our ML baselines for each task in terms of both makespan and runtime.
>
> ### **Algorithmic Improvement via DAG-based optimization.**
>
> As we already discussed in **[Reviewer p2nE, Q2]**, our node priority does not have to satisfy the DAG’s precedence constraints, and list scheduling handles this by using the concept of *ready nodes*. Therefore, our method already optimizes schedules based on DAGs.

---

> ### Author Response · Authors · 2022-11-24
> **Gentle reminder for Reviewer p2nE**
>
> We would like to kindly ask Reviewer p2nE if our response was sufficient to answer your questions and resolve your concerns. Since  Discussion Stage 2 will last until December 12th (from Author Guide of ICLR 2023), we would like to hear your feedback at your earliest convenience and will be willing to resolve any remaining concerns. We truly appreciate your effort to review our work!

---

> > ### Comment · Reviewer_p2nE · 2022-11-26
> > **thanks**
> >
> > Thanks for the reply! I do appreciate your effort in the rebuttal stage, and again I find the empirical results to be solid for this specific problem. Overall I would keep my evaluation where this paper could be a borderline one and I'm a bit negative on its novelty.

---

> > > ### Author Response · Authors · 2022-11-30
> > > **Comment to Reviewer p2nE**
> > >
> > > We appreciate Reviewer p2nE's response. We answered this in 2nd response to Reviewer p2nE below.

---

### Author Response · Authors · 2022-11-16
**General response to reviewers**

We would like to appreciate all reviewers for their valuable comments. We separated reviewers' concerns into general and individual concerns; for general concerns across multiple reviewers, we try to address those via **general responses**. We made comments for each reviewer to address individual concerns. Below, we briefly summarize our revision:

1. We try our best to address Reviewer p2nE and HjB3's concerns on the novelty of our work. Specifically, we added ablation studies with all of our scheduling tasks that show the superiority of our method compared to the existing work on Topoformer by Gagrani et al. [1]. In summary, we empirically show that using logit norm regularization and standardized costs in our submission **always** leads to better performance compared to that of [1] **over all scheduling tasks we consider in our work**. Notably, such gain becomes more significant in the following tasks, which lets us claim that *our technical contribution is not incremental*:
    - In DAG scheduling with TPC-H dataset, one-shot neural schedulers trained w/o our techniques are always outperformed by our neural baseline, PPO-BiHyb [16], which means that **our techniques are necessary to beat our neural baseline**.
    - In the task of scheduling for synthetic computation graphs, one-shot neural schedulers trained w/o our techniques are in most cases beaten by our simple non-neural baseline, MOPNR, which means that **our techniques are crucial to achieve consistently better performance than that of our non-neural baseline.**

2. For Reviewer zenr's request to elaborate practicality of our work, **we (1) provide the time and epochs required for our method to converge and (2) summarize the number of training graphs** that were in our submission. We believe the provide information is beneficial to the readers who are interested in our neural scheduler.

3. For other individual concerns and questions that are not generally raised, we added comments to every concern and question.

4. In our revised manuscript, we clarify our contributions in the introduction, add the ablation studies in Appendix G, and describe the practical details in Appendix F.

Again, we appreciate the valuable feedback of the reviewers and hope our responses can address their concerns and clarify the contributions of our work.

---

> ### Author Response · Authors · 2022-11-16
> **[General Response 1 (for Reviewer p2nE, HjB3)] Novelty of our work**
>
> Although we agree that there are many RL-based heuristics for combinatorial optimizations (CO), most of them are aimed at the variants of TSP which are hugely different from scheduling problems [2]. Historically, scheduling problems themselves are sufficiently complicated and regarded as major topics in operations research. **Our contribution is devising a ML-scheduler that solves a variety of scheduling problems while qualitatively performing the best (compared to our baselines) and requiring the shortest runtime among ML schedulers.** We believe our contribution is significant since improving the practicality of ML itself is impactful and makes existing fundamental ML ideas more meaningful, which is why ICLR is open to topics on ML applications in various fields. Also, regarding Reviewer p2nE’s comments, we would like to emphasize the following contributions:
>
> - Reviewer p2nE mentioned that using the Gumbel-top-k is not new. However, if we restrict our problems of interest to scheduling, **our work is the first one applying the Gumbel-top-k trick to a wide range of scheduling problems.** To support this claim, we search over *every paper* citing [3] (the very first paper introducing the Gumbel-top-k trick to ML community). Among 105 papers, only 5 papers [4,5,6,7,8] solve well-known CO problems like TSP, CVRP, etc, and **none of those works has targeted scheduling problems**. Also, they did not use the Gumbel-top-k trick for priority sampling policy as we did; [4,5,6] used stochastic beam search method (proposed by [3]), and [7,8] used the Gumbel-top-k for gradient variance reduction.
>
> - Although we did use Topoformer encoder [1] and REINFORCE [9], we propose techniques to improve the performance (policy-gradient baseline, logit norm regularization), which is crucial to the performance. We will discuss this in **[General Response 2]** with additional experiments in our revision.

---

> ### Author Response · Authors · 2022-11-16
> **[General Response 2-1 (for all Reviewers)] Comparison with Gagrani et al. [1]**
>
> Our method directly uses Topoformer encoder from [1] to get better abstraction on DAGs, and *there is no trade-off between performance and speed* as the concern raised by Reviewer p2nE. We summarize the difference between [1] and our method below:
> - *Our method*:
>   1. Aim to solve scheduling problems
>   2. Perturb logits (via i.i.d. Gumbel noise) to sample node priorities (not necessarily satisfying precedence constraints) and convert the priorities into schedules via list scheduling.
>   3. While using REINFORCE, cost standardization is used.
>   4. Logits are regularized, i.e., its L2 norm is minimized together with REINFORCE loss.
> - *Gagrani et al. [1]’s method*:
>   1. Aim to solve peak memory minimization (which is *not* a scheduling problem).
>   2. Logits are used to sample sequences *satisfying* the precedence constraints on DAGs.
>   3. While using REINFORCE, greedy baseline is used, motivated by [15].
>   4. Logits are standardized.
>
> For items 3 and 4, our motivations are described as follows. REINFORCE with greedy baseline in [1] uses the policy gradient
> $
> \mathbb{E}\_{\vec{V}\sim\pi\_\theta(\cdot|G)} \nabla\_\theta\log \pi\_\theta (\vec{V}|G) [C(\vec{V};G) - C(\pi\_{\theta’}(G);G)]
> $ for a fixed $G$, where $\pi\_{\theta’}$ is the greedy policy, and its parameter $\theta’$ is copied from $\theta$ when the policy $\pi\_{\theta}$ with $\theta$ shows the best performance during training (normally evaluated at the end of multiple epochs). The problems of greedy baseline are: (1) it may become unstable if cost scales differ too much across training graphs, (2) it slows down training since further forward computation with $\pi\_{\theta’}$ is required, and (3) it requires additionally memory to store $\theta’$, **all of which can be resolved by the cost standardization technique that we used in our paper**. Also, if we use logit standardization used in [1], we empirically observed performance degradation, which we believe is due to logit standardization restricting the representation power of the model and motivated us to come up with logit norm regularization (we presented a simple binary-random-variable example in Appendix A in our submission).
>
> To support our claim and prove the effectiveness of our technical improvement to Reviewers, we pursue further ablation studies for our algorithm with all tasks in our submission. Specifically, **we compare our method with [1]’s method using both greedy baseline and logit standardization** while maintaining the priority sampling via Gumbel and list scheduling (this is necessary since we have to fairly compare performances in scheduling domains). Then, we train the model for all scheduling tasks we presented in our submission; JSSP, DAG scheduling, real and synthetic computation graphs. We report tabular results below as well as in our revision:
>
> - **JSSP.** Our method is shown to perform better than [1]’s method. **Our method always outperform [1]’s method when they are in the same mode and the same task. Runtimes for both methods are comparable with each other.** Although we didn’t specify, [1]’s method can outperform our neural and non-neural baselines in JSSP (Please see Appendix G for more details).
> |                                     | (25, 20) |            | (25, 30) |            | (50, 20) |            |
> |-------------------------------------|----------|------------|----------|------------|----------|------------|
> |                                     | Makespan | Time (sec) | Makespan | Time (sec) | Makespan | Time (sec) |
> |                                     |          |            |          |            |          |            |
> | Greedy ([1]'s method)          | 2077.6   | 0.022      | 2610.86  | 0.029      | 3129.16  | 0.056      |
> | Sampling (16) ([1]'s method)   | 2004.78  | 0.055      | 2493.38  | 0.083      | 3055.2   | 0.142      |
> | Sampling (64)  ([1]'s method)  | 1979.86  | 0.129      | 2467     | 0.291      | 3037.38  | 0.484      |
> | Sampling (256)  ([1]'s method) | 1957.76  | 0.512      | 2446.04  | 0.844      | 3015.78  | 1.568      |
> |                                     |          |            |          |            |          |            |
> | Greedy (ours)                       | 2032.7   | 0.021      | 2512.4   | 0.031      | 3108.56  | 0.049      |
> | Sampling (16) (ours)                | 1970.98  | 0.054      | 2452.64  | 0.085      | 3032.44  | 0.138      |
> | Sampling (64)  (ours)               | 1948.76  | 0.127      | 2427.3   | 0.294      | 3009.08  | 0.469      |
> | Sampling (256)  (ours)              | **1932.42**  | 0.514      | **2411.68**  | 0.909      | **2997.1**   | 1.527      |
> |                                     |          |            |          |            |          |            |

---

> ### Author Response · Authors · 2022-11-16
> **[General Response 2-2 (for all Reviewers)] Comparison with Gagrani et al. [1]**
>
>
> - **DAG scheduling (TPC dataset).** Overall, **our method outperforms [1]’s method.** **Runtimes for both methods are comparable with each other**. For TPC dataset, our method can be regarded to make significant improvement due to the following observations:
>   1. **Sampling (256) mode of [1] (that shows the lowest makespan among [1]’s runs) is always outperformed by Greedy mode of our method (which is the highest makespan among our methods).**
>   2. **Our neural baseline, PPO-BiHyb [16], always outperforms [1]’s method for all TPC tasks.**
> |                                | TPC-H-50 |            | TPC-H-100 |            | TPC-H-150 |            |
> |--------------------------------|----------|------------|-----------|------------|-----------|------------|
> |                                | Makespan | Time (sec) | Makespan  | Time (sec) | Makespan  | Time (sec) |
> |                                |          |            |           |            |           |            |
> | PPO-BiHyb                      | *8905.4*   | 66.484     | *15192.2*   | 149.215    | *22371.2*   | 571.424    |
> |                                |          |            |           |            |           |            |
> | Greedy ([1]'s method)          | 9300.4   | 0.219      | 16185.5   | 0.139      | 23788.9   | 0.143      |
> | Sampling (16) ([1]'s method)   | 9079.9   | 0.322      | 15974.1   | 0.34       | 23477.6   | 0.425      |
> | Sampling (64)  ([1]'s method)  | 9037.3   | 0.467      | 15804.1   | 0.748      | 23412.2   | 1.233      |
> | Sampling (256)  ([1]'s method) | 8976.2   | 0.959      | 15684.9   | 2.338      | 23264.4   | 4.419      |
> |                                |          |            |           |            |           |            |
> | Greedy (ours)                  | 8845.6   | 0.057      | 14981.2   | 0.1        | 22332.7   | 0.259      |
> | Sampling (16) (ours)           | 8782.4   | 0.114      | 14972     | 0.287      | 22330.2   | 0.674      |
> | Sampling (64)  (ours)          | 8742.5   | 0.216      | 14968.1   | 0.699      | 22323     | 1.856      |
> | Sampling (256)  (ours)         | **8694.4**   | 0.54       | **14964.7**   | 2.27       | **22320.8**   | 6.485      |
> |                                |          |            |           |            |           |            |

---

> ### Author Response · Authors · 2022-11-16
> **[General Response 2-3 (for all Reviewers)] Comparison with Gagrani et al. [1]**
>
>
> - **Computation graph scheduling (synthetic).** Overall, **our method outperforms [1]’s method.** **Runtimes for both methods are comparable with each other**. For synthetic computation graphs, the following observations make our method more outstanding:
>   1. **Sampling (256) mode of [1] (that shows the lowest makespan among [1]’s runs) is always outperformed by Greedy mode of our method (which is the highest makespan among our methods).**
>   2. **Our non-neural baseline, MOPNR, always outperforms [1]’s Greedy mode and some of sampling modes.**
> |                                | Layered Graph |            | Erdos-Renyi |            | Stock. Block Model |            |
> |--------------------------------|---------------|------------|-------------|------------|--------------------|------------|
> |                                | SpeedUp       | Time (sec) | SpeedUp     | Time (sec) | SpeedUp            | Time (sec) |
> |                                |               |            |             |            |                    |            |
> | MOPNR                          | *4.745*         | 0.075      | *5.112*       | 0.068      | *4.761*              | 0.068      |
> |                                |               |            |             |            |                    |            |
> | Greedy ([1]'s method)          | 4.597         | 0.051      | 4.871       | 0.047      | 4.657              | 0.06       |
> | Sampling (16) ([1]'s method)   | 4.706         | 0.196      | 4.989       | 0.219      | 4.792              | 0.217      |
> | Sampling (64)  ([1]'s method)  | 4.74          | 0.515      | 5.028       | 0.529      | 4.833              | 0.536      |
> | Sampling (256)  ([1]'s method) | 4.767         | 1.73       | 5.056       | 1.716      | 4.863              | 1.775      |
> |                                |               |            |             |            |                    |            |
> | Greedy (ours)                  | 4.819         | 0.094      | 5.194       | 0.046      | 4.866              | 0.043      |
> | Sampling (16) (ours)           | 4.848         | 0.311      | 5.211       | 0.214      | 4.916              | 0.198      |
> | Sampling (64)  (ours)          | 4.872         | 0.75       | 5.227       | 0.542      | 4.944              | 0.477      |
> | Sampling (256)  (ours)         | **4.889**         | 2.418      | **5.239**       | 1.799      | **4.965**              | 1.533      |
> |                                |               |            |             |            |                    |            |
>
> - **Computation graph scheduling (real-world).** Our method outperforms [1]’s method and shows comparable runtimes for all tasks.
> |                                | 200-500 Node Graphs |            | 500-700 Node Graphs |            | 700-1000 Node Graphs |            |
> |--------------------------------|---------------------|------------|---------------------|------------|----------------------|------------|
> |                                | SpeedUp             | Time (sec) | SpeedUp             | Time (sec) | SpeedUp              | Time (sec) |
> |                                |                     |            |                     |            |                      |            |
> | Greedy ([1]'s method)          | 3.227               | 0.196      | 3.114               | 0.028      | 2.822                | 0.045      |
> | Sampling (16) ([1]'s method)   | 3.248               | 0.243      | 3.151               | 0.153      | 2.843                | 0.238      |
> | Sampling (64)  ([1]'s method)  | 3.254               | 0.324      | 3.179               | 0.404      | 2.847                | 0.656      |
> | Sampling (256)  ([1]'s method) | 3.258               | 0.644      | 3.192               | 1.317      | 2.853                | 2.205      |
> |                                |                     |            |                     |            |                      |            |
> | Greedy (ours)                  | 3.245               | 0.152      | 3.131               | 0.098      | 2.846                | 0.06       |
> | Sampling (16) (ours)           | 3.271               | 0.192      | 3.188               | 0.245      | 2.848                | 0.23       |
> | Sampling (64)  (ours)          | 3.278               | 0.263      | 3.199               | 0.456      | 2.856                | 0.606      |
> | Sampling (256)  (ours)         | **3.286**               | 0.595      | **3.207**               | 1.309      | **2.86**                 | 2.001      |
> |                                |                     |            |                     |            |                      |            |

---

> ### Author Response · Authors · 2022-11-16
> **References**
>
> [1] Mukul Gagrani, Corrado Rainone, Yang Yang, Harris Teague, Wonseok Jeon, Herke Van Hoof, Weiliang Will Zeng, Piero Zappi, Christopher Lott, Roberto Bondesan. Neural Topological Ordering for Computation Graphs. NeurIPS 2022.
>
> [2] Yoshua Bengio, Andrea lodi, and Antoine Prouvost. Machine Learning for Combinatorial Optimization: a Methodological Tour d’Horizon. arXiv 2020.
>
> [3] Wouter Kool, Herke van Hoof, Max Welling. Stochastic beams and where to find them: The Gumbel-top-k trick for sampling sequences without replacement. ICML 2019.
>
> [4] Kensen Shi, David Bieber, and Charles Sutton. Incremental sampling without replacement for sequence models. ICML 2020.
>
> [5] Liang Xin, Wen Song, Zhiguang Cao, and Jie Zhang. Multi-decoder attention model with embedding glimpse for solving vehicle routing problems. AAAI 2021.
>
> [6] Eldan Cohen and J. Christopher Beck. Heavy-tails and randomized restarting beam search in goal-oriented neural sequence decoding. CPAIOR 2021.
>
> [7] Wouter Kool, Herke van Hoof, Max Welling. Estimating gradients for discrete random variables by sampling without replacement. ICLR 2020.
>
> [8] Wouter Kool, Herke van Hoof, Max Welling. Buy 4 REINFORCE samples, get a baseline for free!. ICLR 2019 Workshop.
>
> [9] Ronald J Williams, Simple statistical gradient-following algorithms for connectionist reinforcement learning. Machine Learning 1992.
>
> [10] Ronald L. Graham. Bounds on multiprocessing timing anomalies. SIAM journal on Applied Mathematics 1969.
>
> [11] Eric Jang, Gu Shixiang, and Ben Poole. Categorical reparametrization with Gumble-softmax. ICLR 2017.
>
> [12] Chris J. Maddison, Andriy Mnih, and Yee Whye Teh. The concrete distribution: A continuous relaxation of discrete random variables. ICLR 2017.
>
> [13] John Schulman, Filip Wolski, Prafulla Dhariwal, Alec Radford, and Oleg Klimov. Proximal policy optimization algorithms. ArXiv 2017.
>
> [14] John Schulman, Philipp Moritz, Sergey Levine, Michael Jordan, and Pieter Abbeel. High-dimensional continuous control using generalized advantage estimation. Arxiv 2015.
>
> [15] Wouter Kool, Herke van Hoof, and Max Welling. Attention, learn to solve routing problems!. ICLR 2019.
>
> [16] Runzhong Wang, Zhigang Hua, Gan Liu, Jiayi Zhang, Junchi Yan, Feng Qi, Shuang Yang, Jun Zhou, and Xiaokang Yang. A bi-level framework for learning to solve combinatorial optimization on graphs. NeurIPS 2021.
>
> [17] Liang Xin, Wen Song, Zhiguang Cao, and Jie Zhang. NeuroLKH: Combining deep learning model with Lin-Kernighan-Helsgaun heuristic for solving the traveling salesman problem. NeurIPS 2021
>
> [18] Aditya Paliwal, Felix Gimeno, Vinod Nair, Yujia Li, Miles Lubin, Pushmeet Kohli, and Oriol Vinyals. Reinforced genetic algorithm learning for optimizing computation graphs. ICLR 2020.
>
> [19] Cong Zhang, Wen Song, Zhiguang Cao, Jie Zhang, Puay Siew Tan, and Xu Chi. Learning to dispatch for job shop scheduling via deep reinforcement learning. NeurIPS 2020

---

### Decision · Program_Chairs · 2023-01-20

**Decision:**

Accept: poster

**Justification For Why Not Higher Score:**

The novelty of the work is slightly limited and it is not a huge breakthrough.

**Justification For Why Not Lower Score:**

I had a similar view as the reviewer zenr. I acknowledge authors' efforts during rebutalls (many concerns were addressed), though reviewers keep the scores unchanged. I think the rejecting a publication because it is "sort of interesting but not a breakthrough" is unreasonable.

**Metareview: Summary, Strengths And Weaknesses:**

Summary:
This paper considers list scheduling on DAGs using one-shot neural networks. The proposed approach is much faster than neural baselines and achieves good empirical performance in a few tasks.

Strength:
The paper is well written. The proposed techniques achieve solid empirical results. During rebuttals, most of the concerns were addressed by the authors.

Weakness:
The main concern from reviewers is the novelty of the work as it is by combining a few methods (Topoformer, REINFORCE, Gumbel-Top-k on a new task).

**Note From Pc:**

if the above contains the word "oral" or "spotlight" please see: "oral" presentation means -> notable-top-5% and "spotlight" means -> notable-top-25%. As stated in our emails, we are disassociating presentation type from AC recommendations